# SutraNets: Sub-series Autoregressive Networks for Long-Sequence, Probabilistic Forecasting

**Shane Bergsma    Timothy Zeyl    Lei Guo**
Huawei Cloud, Alkaid Lab Canada
{shane.bergsma,timothy.zeyl,leiguo}@huawei.com

## Abstract

We propose SutraNets, a novel method for neural probabilistic forecasting of long-sequence time series. SutraNets use an autoregressive generative model to factorize the likelihood of long sequences into products of conditional probabilities. When generating long sequences, most autoregressive approaches suffer from harmful error accumulation, as well as challenges in modeling long-distance dependencies. SutraNets treat long, univariate prediction as *multivariate* prediction over lower-frequency *sub-series*. Autoregression proceeds across time *and* across sub-series in order to ensure coherent multivariate (and, hence, high-frequency univariate) outputs. Since sub-series can be generated using fewer steps, SutraNets effectively reduce error accumulation and signal path distances. We find SutraNets to significantly improve forecasting accuracy over competitive alternatives on six real-world datasets, including when we vary the number of sub-series and scale up the depth and width of the underlying sequence models.

## 1   Introduction

As a cloud provider, we rely on long-range forecasts to allocate resources and plan capacity. In particular, we need to predict the *fine-grained* pattern of demand (e.g., hourly) for a large number of products, over a long time horizon (e.g., months). Since purchase decisions are large-scale (e.g., supplying regions with 100K machines), and made weeks-to-months before delivery, cloud providers have a profound need for methods that "capture the uncertainty of the future" [43]. Fine granularity, long histories, long forecast horizons, and *quantification of uncertainty* can be formalized as the long-sequence (many-step) *probabilistic* forecasting problem.

Recent work applies Transformers [82] to long-sequence forecasting. In order to circumvent the Transformer's quadratically-scaling *attention* mechanism, long sequences necessitate trading off increased signal path for reduced computational complexity [44, 92, 87, 47]. Unfortunately, the effectiveness and even validity of these approaches have been questioned [90, 32]. Most of these approaches do not provide probabilistic outputs (rather best-guess point predictions), and all condition on fixed-size windows, typically of very short duration. Indeed, many recent papers on "long-term" forecasting [87, 86, 70, 93] condition on a maximum of 96 inputs — only 4 days of history at 1-hour granularity. Such restricted context is a serious limitation given that many time series (including in standard evaluation datasets based on electricity and traffic) have strong *weekly* seasonality.

Our production forecasting system, like many in industrial settings [65, 52, 72] and cloud services [69], is based on RNNs. The RNN's recurrent architecture forces the compression and consolidation of long-range information, and RNNs work well on smaller datasets as they are not as "data hungry" as Transformers [16]. RNNs can theoretically condition on context of any length; in this paper, we condition RNN forecasts on length-2016 contexts without difficulties in scaling.

RNNs are often used to generate probabilistic forecasts via an autoregressive approach [21, 65, 60, 7], whereby the probability over future timesteps is factorized into a product of one-step-ahead

37th Conference on Neural Information Processing Systems (NeurIPS 2023).

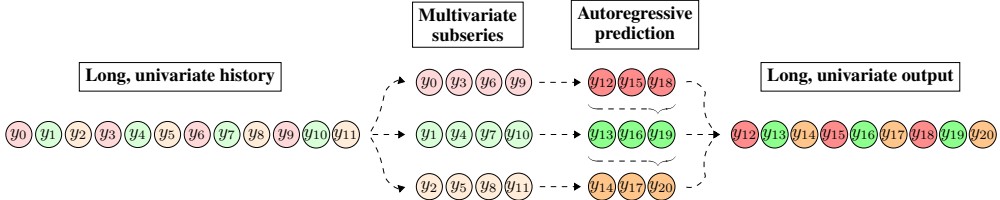

Figure 1: SutraNets: a long, univariate history is converted to lower-frequency sub-series. Multivariate autoregressive prediction, across time and sub-series, ensures coherent univariate output samples.

conditionals [6, 28]. When generating many steps, however, two key problems arise: (1) a discrepancy grows between training, where we condition on true previous values, versus inference, where we condition on sampled values (the *discrepancy* problem), and (2) long-term dependencies may be challenging to exploit (the *signal path* problem). The discrepancy problem leads to error accumulation during inference [5], but also a kind of *informational asymmetry*: because highly-informative true values are available during training, the model may ignore other useful info, e.g., seasonally-lagged inputs provided as extra features. The signal path problem, meanwhile, is especially acute for RNNs, where distant information must be preserved over many steps in the RNN hidden state.

We propose SutraNets,[1] a general method for probabilistic prediction of long time series. We focus on the application of SutraNets to RNN-based forecasting, where they offer major benefits. SutraNets transform a length-$N$ series into $K$ sub-series of length $N/K$. Sub-series forecasts are generated autoregressively, sequentially conditional on each other, enabling coherent outputs (Fig. 1). SutraNets address both the discrepancy problem (by forcing the network to take larger *generative strides*, i.e., predicting without access to immediately-preceding true values) and the signal path problem (by reducing the distance between historical and output values by a factor of $K$). Training of SutraNets *decomposes* over sub-series; i.e., sub-series RNNs can be trained in *parallel*, enabling a $K$-fold improvement in training parallelism over standard RNNs.

Experimentally, SutraNets demonstrate superior forecasting across six datasets, improving accuracy by an average of 15% over C2FAR [7] and over strong seasonality-aware baselines. We evaluate a variety of other strategies for improving long-sequence forecasting, including lags, input dropout, deeper and wider networks, and multi-rate hierarchical approaches. Through these evaluations, we gain insights into how to succeed at long-sequence generation, in time series and beyond.

## 2 Background and related work

**Probabilistic autoregressive forecasting**    Autoregressive forecasting models typically combine a backbone sequence model with a technique to estimate an output probability distribution at each step. In this paper, we apply SutraNets to C2FAR [7], a state-of-the-art distribution estimator that was shown to improve over DeepAR [65], DeepAR-binned [60], SQF-RNN [21], and IQN-RNN [27]. SutraNets can serve as the sequential backbone for any of these estimators, as well as for other recent approaches such as the denoising diffusion models [62], or conditioned normalizing flows [63].

**Multi-rate forecasting**    It is sometimes acceptable to *aggregate* high-frequency series or event data to a lower frequency prior to sequence modeling, e.g., aggregating patient events over 6-month intervals [48], or network traffic over 10 subframes [79]. When high-frequency predictions are required, lower-frequency forecasts can provide guidance. Techniques for *temporal disaggregation* convert low-frequency series to higher frequencies via related high-frequency *indicator* series [13, 67]. Athanasopoulos et al. [2] simultaneously forecast at multiple frequencies and then reconcile the predictions. Such methods do not provide uncertainty estimates nor are they based on neural networks.

Scaleformer [70] aggregates series, forecasts at the lower frequency, and upsamples the lower-freq predictions in order to provide additional inputs for the original forecast model. We evaluate a similar approach, Low2HighFreq (§3), as a baseline. Other neural approaches (in forecasting and beyond) implicitly encourage learning of lower-frequency dynamics through hierarchical attention [20, 47, 16],

---

[1]In Sanskrit, a *sūtra* is a thread or "that which like a thread runs through or holds everything together" [85]. A SutraNet prediction is woven from threads of sub-series predictions; Fig. 2 shows different stitching patterns.

pooling blocks [92, 87, 49, 93], and deeply-stacked [29, 81] and multi-rate [68, 18, 38, 73, 14, 11] recurrence layers. Unlike these approaches, SutraNets *explicitly* generate *low-frequency* sub-series conditional on other low-frequency sub-series, without directly forecasting at a high frequency at all.

**Reducing training/inference discrepancy**   If we know *a priori* how far to forecast, it is possible to avoid error accumulation by directly forecasting all horizons in one step [76], or via non-autoregressive decoding [20]. Such systems usually generate point predictions [92, 87, 9], although predicting certain fixed forecast quantiles is also possible [84, 20]. ProTran [77] is a probabilistic but non-autoregressive approach. Without modeling local dependencies among outputs, such models prevent generation of *coherent* samples, and preclude computing robust likelihoods for *given* output sequences — where error accumulation is not an issue (e.g., for anomaly detection, missing value interpolation, compression, etc.). SutraNets are more in-step with flexible autoregressive models like GPT3 [8].

For autoregressive models, *scheduled sampling* [5] aims to handle errors better during inference by sampling (incorrect) inputs during *training*. This approach was effective in tasks such as human motion synthesis [45], but results have been mixed in autoregressive forecasting [65, 66]. Noting that scheduled sampling produces a biased estimator, Lamb et al. [40] instead used adversarial techniques to reduce training/inference discrepancy. Wu et al. [88] applied this method to forecasting, using a GAN [26] with an autoregressive sparse transformer. However, adversarial training showed insignificant gains over the sparse transformer alone on 7-day electricity and traffic predictions.

Like scheduled sampling, *dropout* [74] is a kind of stochastic regularization that prevents overfitting by injecting noise into networks. Dropout rates of around 20% are commonly used in input layers of autoencoders and feed-forward nets [3, 74]. Dropout is also common in autoregressive models [82, 61], where, in theory, it may encourage networks to rely less on (now noisy) previous values [56].

**Modeling long-term dependencies**   When forecasting long series, seasonally *lagged* values — historical values from, e.g., one year ago — can be used as extra features in the sequence model, acting as a kind of residual input [31]. A recent competition featured long time series (e.g., 700 days of history) [1]; the winning model found lag to be more effective than fixed attention weights [75]. Some researchers have even declared that, when it comes to attention, "lag is all you need" [19].

Learned *attention* can capture long-range dependencies [82], but is infeasible for very long sequences due to memory and time complexity quadratic in sequence length. Complexity also scales quadratically for forecasting via fully-connected layers [57, 9]. Recent work has explored sparsified or hierarchical attention in forecasting [44, 92, 47, 87]. TimesNet [86] transforms univariate time series into 2D tensors of multiple periods, which are then processed via Inception-style 2D kernels to generate point predictions. As mentioned earlier, most of these systems fail to condition on more than very short windows — a few days of hourly data. Ultimately, we are interested in *probabilistic* forecasting using *years* of fine-grained historical data. It is unclear whether lag is really all we need, or even whether seasonal-naive baselines [34] are already more effective than Transformers [32].

Unlike Transformers, RNNs can be applied to long sequences *out-of-the-box*. However, since RNN signal path is linear in sequence length, mechanisms are required to preserve long-range info, such as the special gated units of LSTMs [33]. Multi-dimensional RNNs [30] propagate distant info via row and column connections across images [78, 81]. Didolkar et al. [16] combine RNNs for long-range signals with Transformers that attend over shorter chunks. SutraNets, in contrast, do not require architectural changes *within* RNNs; indeed, they work with RNNs or Transformers. In SutraNets, low-frequency information is both *implicitly* captured in network state and *explicitly* and autoregressively communicated through generated low-frequency forecasts.

**Comparison to other RNNs**   Table 1 compares SutraNets to prior RNNs (cf. Table 1 in [82]). PatchTST [55] groups consecutive values into input patches. While patching was proposed as a method to reduce the attentional complexity of Transformers, patching was also recently used with RNNs in SegRNN [46], where it effectively reduces the signal path by a factor of $K$ (taking in $K$ inputs each step — like SutraNets — but with $K$ times fewer overall input steps). However, neither PatchTST nor SegRNN provides a mechanism to probabilistically *generate* patches, and thus generate coherent *probabilistic* outputs. Dilated LSTMs [83] and Dilated RNNs [10] with minimum dilation amounts >1 do facilitate error reduction and parallel training, but at the cost of sacrificing coherency, as the model is then "equivalent to multiple shared-weight networks, each working on partial inputs"[10, §4.4]. With such DilatedRNNs, if the time series were to spike to a high level at

Table 1: Comparison of SutraNets to prior RNNs. Like prior methods, SutraNets reduce signal path, but also improve error accumulation (via a larger generative stride), while enabling greater training parallelism (via fewer sequential operations), without sacrificing coherent *probabilistic* output.

| RNN method | Meaning of K | Signal path | Generative max stride | Sequential operations | Coherent outputs |
|---|---|---|---|---|---|
| Standard RNN, e.g., LSTM [33] | N/A | $\mathcal{O}(N)$ | $\mathcal{O}(1)$ | $\mathcal{O}(N)$ | Yes |
| Skips [91], lags/residuals [31] | Skip/lag amount | $\mathcal{O}(N/K)$ | $\mathcal{O}(1)$ | $\mathcal{O}(N)$ | Yes |
| PatchTST [55], SegRNN [46] | Patch size/stride | $\mathcal{O}(N/K)$ | N/A | $\mathcal{O}(N/K)$ | N/A |
| DilatedRNN [10], dilations $C\ldots K$ | Largest dilation | $\mathcal{O}(N/K)$ | $\mathcal{O}(C)$ | $\mathcal{O}(N/C)$ | If $C$=1 |
| SutraNets | Num. sub-series | $\mathcal{O}(N/K)$ | $\mathcal{O}(K)$ | $\mathcal{O}(N/K)$ | Yes |

the very final conditioning input, only a subset of the outputs would be generated conditional on this spike, resulting in highly incoherent output. This can be mitigated for point predictions by adding a final *fusion layer* [10], but such an approach is not compatible with probabilistic forecasts.

Another related approach is the Subscale WaveRNN model for audio generation [35] (conditioned on input text). For Subscale WaveRNN, the motivation is not to reduce signal path nor improve error accumulation, but to have a $K$-fold increase in inference parallelism when generating very, very long audio sequences. Generation proceeds somewhat similar to our regular, non-alternating version (§3) but conditioning on past and some *future* values, and with a shared neural network for all sub-series.

## 3   SutraNets: Sub-series autoregressive networks

Let $y_t \in \mathbb{R}$ be the value of a time series at time $t$, and $\mathbf{x}_t$ be a vector of time-varying features or *covariates*. Let $y_{i:j}$ denote sequence $y_i \ldots y_j$. We seek a model of the distribution of $N$ future values of $y_t$ (the *prediction range*) given $T$ historical values (the *conditioning range*). We can formulate this conditional distribution as an autoregressive (AR) generative model as in [28]:

$$p(y_{T+1:T+N}|y_{1:T}, \mathbf{x}_{1:T+N}) = \prod_{t=T+1}^{T+N} p(y_t|y_{1:t-1}, \mathbf{x}_{1:T+N}) \tag{1}$$

Following DeepAR [65], forecasting approaches have modeled the one-step-ahead distributions using RNNs [21, 65, 60, 7] or Transformers [44]. In these approaches, a global sequence model is trained by slicing many training series into many *windows*, i.e., conditioning+prediction ranges at different start points, and normalizing the windows using their conditioning ranges. Model parameters are fit via gradient descent, minimizing NLL of prediction-range outputs. At inference time, forecasts for a given conditioning range are generated by sampling $\hat{y}_{T+1} \ldots \hat{y}_{T+N}$ sequentially, autoregressively conditioning on samples generated at previous timesteps. By repeating this procedure many times, a Monte Carlo estimate of Eq. (1) is obtained, from which desired forecast quantiles can be derived.

SutraNets consider each univariate window to be comprised of an ordered collection of $K$ lower-frequency sub-series, each $1/K$ of the original length (Fig. 1). The $k$th sub-series is obtained by selecting every $K$th value in the original window, each sub-series starting at a unique offset in $1 \ldots K$.[2] Let $y^k$ be the $k$th sub-series. Let $\tilde{N} = N/K$ and $\tilde{T} = T/K$ be prediction and conditioning lengths corresponding to each sub-series, and let $<k$ and $>k$ denote indices from $1 \ldots K$ that are less than and greater than $k$. Through another application of the chain rule, now across sub-series, Eq. (1) becomes:

$$p(y_{T+1:T+N}|y_{1:T}, \mathbf{x}_{1:T+N}) = \prod_{\tilde{t}=\tilde{T}+1}^{\tilde{T}+\tilde{N}} \prod_{k=1}^{K} p(y_{\tilde{t}}^k|y_{1:\tilde{t}}^{<k}, y_{1:\tilde{t}-1}^k, y_{1:\tilde{t}-1}^{>k}, \mathbf{x}_{1:T+N}) \tag{2}$$

where $\tilde{t}$ iterates over timesteps in each sub-series. Log-likelihoods of sub-series prediction ranges can be summed to compute overall NLL, which is minimized during SutraNet training. Rather than modeling these conditionals using a single RNN, SutraNets use a separate RNN for each sub-series.

---

[2]Since the $k$th sub-series is assigned after slicing training/testing windows, it will vary in terms of which offset it corresponds to in the *original* time series, e.g., it may start at 1pm, 2pm, 3pm, etc., in different windows.

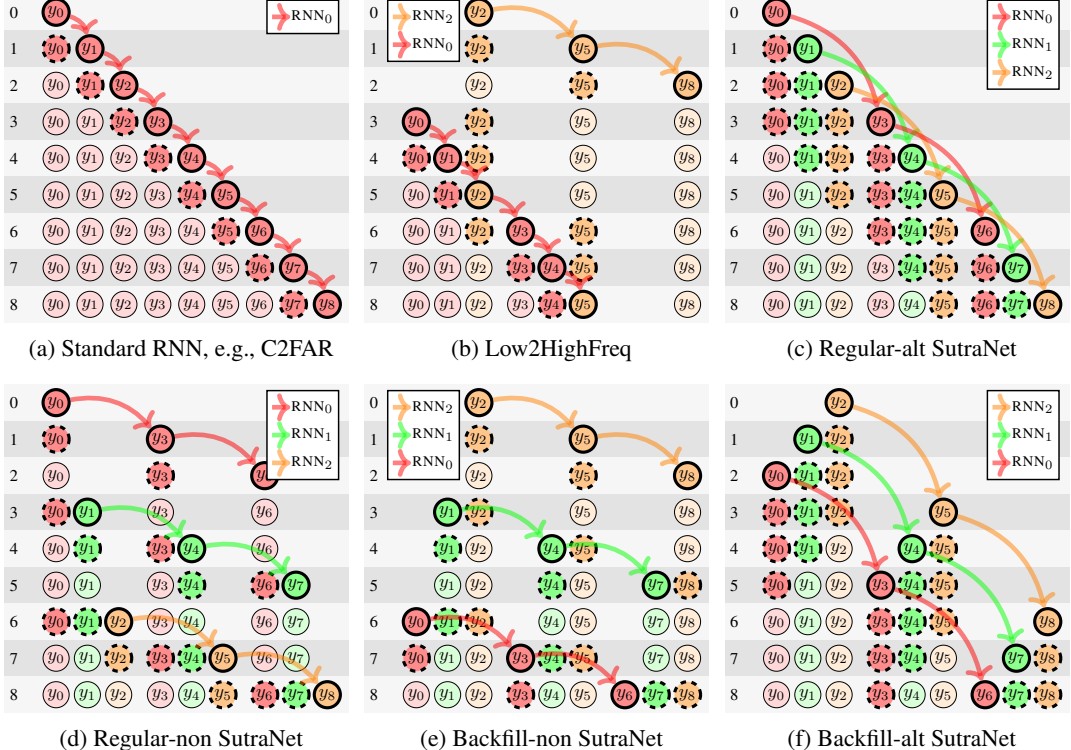

(a) Standard RNN, e.g., C2FAR     (b) Low2HighFreq     (c) Regular-alt SutraNet

(d) Regular-non SutraNet     (e) Backfill-non SutraNet     (f) Backfill-alt SutraNet

Figure 2: Ordering of generative steps in SutraNets. One output value (bold border) is generated in each row, conditional on (1) feature nodes (dashed border) *in that row*, and on (2) state from previous steps ($\Longrightarrow$ arrow connections). SutraNet RNNs (c-f) use features generated in earlier steps by both themselves and other RNNs; these values can even be from future timesteps (backfill cases).

Each RNN has its own parameters and hidden state, but SutraNets are trained and tuned collectively as a single network. The $k$th RNN determines the $k$th conditional probability at each timestep:

$$p(y_{T+1:T+N}|y_{1:T}, \mathbf{x}_{1:T+N}) = \prod_{\tilde{t}=\tilde{T}+1}^{\tilde{T}+\tilde{N}} \prod_{k=1}^{K} p(y_{\tilde{t}}^k|\theta_{\tilde{t}}^k = f(h_{\tilde{t}}^k)) \tag{3}$$

where $h_{\tilde{t}}^k = \text{rnn}^k(h_{\tilde{t}-1}^k, y_{\tilde{t}}^{<k}, y_{\tilde{t}-1}^k, y_{\tilde{t}-1}^{>k}, \mathbf{x}_{\tilde{t}}^k)$ and $f(\cdot)$ is a function mapping the $k$th RNN state to parameters of a parametric output distribution, $p(y_{\tilde{t}}^k|\theta_{\tilde{t}}^k)$.[3] So while prior models condition $y_t$ only on $y_{t-1}$, SutraNets autoregressively condition the generation of $y_{\tilde{t}}^k$ on:

1. $y_{\tilde{t}}^{<k}$: *Current* values (at $\tilde{t}$) of sub-series with indices $< k$ (always)
2. $y_{\tilde{t}-1}^k$: The *previous* value (at $\tilde{t}-1$) of the current sub-series, $k$ (always)
3. $y_{\tilde{t}-1}^{>k}$: *Previous* values (at $\tilde{t}-1$) of sub-series with indices $> k$ (optional)

Conditioning $y_{\tilde{t}}^k$ on $y_{\tilde{t}}^{<k}$ makes the model autoregressive across sub-series; this ensures sub-series are generated conditional on each other, and thus (recombined) samples of the original high-frequency series will be coherent. Conditioning on $y_{\tilde{t}-1}^{>k}$ is optional; when using $y_{\tilde{t}-1}^{>k}$, SutraNets must generate in an *alternating* manner, generating one value for each sub-series at each timestep, $\tilde{t}$ (Fig. 2c). Without $y_{\tilde{t}-1}^{>k}$, SutraNets can generate the complete prediction range of the $k$th sub-series before generating any predictions for $> k$ ones (Fig. 2d); in other words, we can exchange ordering of the products in Eq. (2), remove $y_{\tilde{t}-1}^{>k}$ from the equation, and proceed in a *non-alternating* manner.

---

[3]Note we can input any subset of the covariates to the RNN at each timestep. Here we include those specific to the given timestep of the given sub-series, $\mathbf{x}_{\tilde{t}}^k$, as in practice most covariates are either static across the prediction range (e.g., a product ID), or provide timestep-specific info (e.g., the hour-of-the-day) [65].

SutraNets must also specify how the sub-series index, $k$, relates to the offset in the original series. Figs. 2c and 2d illustrate *regular* ordering, where sub-series $k$ begins at offset $k$ in the original series. An alternative is *backfill* ordering, where sub-series $k$ starts at offset $K$-$k$+1 (Figs. 2e, 2f). We must also specify the number of sub-series, $K$. We pick $K$ so it divides into the primary seasonal period (e.g., $K$=6 for 24-hour seasons), as each sub-series then exhibits seasonality; signal processing techniques can reveal seasonality when it is not known *a priori*. Sub-series with seasonality may be more predictable, which will be especially useful for non-alternating models, as these models generate initial sub-series while conditioning on only $1/K$ of historical values. Success for these models depends on whether there is *redundancy* in the series, or whether values from other sub-series are sufficiently important that non-alternating generation creates a *missing information* problem.

**Reducing training/inference discrepancy, signal path**  Non-alternating and backfill approaches force the network to predict $K$ steps ahead, without access to true previous values. Non-alternating also generates far horizons in fewer steps (e.g., for the $k$=1 sub-series), reducing error accumulation. In contrast, Regular-alt generates in the same order, and conditional on the same values, as standard RNN models (Fig. 2a), and so should suffer the same effects from training/inference discrepancy.

All SutraNets reduce RNN signal path by a factor of $K$, as can be seen by tracing paths between features and outputs in Fig. 2. Note Regular-alt has low signal path but standard discrepancy (as noted above). The relative effectiveness of Regular-alt can thereby provide a diagnostic for whether discrepancy or signal path affects a given forecasting task (§4).

**Form of output distribution and input encoding**  Unlike sampling text from a softmax, forecasting requires a parametric output distribution, $p(y_t^k|\theta_t^k)$ (Eq. (3)), that can account for mixed discrete and continuous outputs, potentially of unbounded dynamic range. C2FAR [7] proposes such an output distribution, via an autoregressive generative model over a hierarchical coarse-to-fine discretization of time series amplitudes (with Pareto-distributed tails). We adopt C2FAR-LSTMs for our sub-series sequence models. We use a 3-level C2F model with 12 bins at each level, discretizing normalized values to a precision of $12^3$ but requiring only $12 \times 3$ output dimensions at each timestep. Previous values from each sub-series and covariate features from other sub-series are also C2F-encoded and provided as inputs at each step. Here the efficiency of C2FAR proves helpful; at each step we can encode $K$ covariate values (i.e., from $K$ other sub-series) using only $12 \times 3 \times K$ input dimensions, as opposed to, e.g., the $12^3 \times K$ dimensions that would be required with equivalent flat binnings [60].

**Low2HighFreq**  Instead of generating all lower-frequency sub-series, we experiment with generating a single low-freq sub-series and conditioning on it within a high-freq model. This Low2HighFreq approach (Fig. 2b) is similar to multi-rate approaches (§2), except here low-freq samples provide not only features, but hard constraints on every corresponding $K$th value in the high-freq output. During inference, when we use Monte Carlo sampling to estimate $p(y_{T+1:T+N}|\ldots)$, we can actually *sample* high-freq values *conditional* on low-freq ones. As such, techniques such as rejection sampling [53] are applicable; we experimented with importance sampling via likelihood weighting (fixing the low-freq values and sampling the rest), but ultimately found the likelihood weights had such high variance that it was more effective to use uniform weights and accept inconsistent estimation.

**Complexity**  Let $H$ be the number of RNN hidden units and $K$ the number of sub-series. The complexity-per-timestep of each SutraNet RNN scales with the sum of the RNN's recurrence, $H \times H$, and projection of (at most) $K$ inputs (from other sub-series) to the RNN hidden state, $K \times H$. Even though SutraNets have $K$ separate RNNs, only one runs at each original timestep (i.e., each SutraNet RNN runs $1/K$ of total timesteps). In consequence, SutraNet complexity-per-timestep scales with $H \times H + K \times H$, whereas standard RNNs scale $H \times H$. In practice, $H >> K$, and we find empirically SutraNets also scale $H \times H$, resulting in similar training/inference speeds and memory requirements compared to standard models, given the same individual LSTM depth and width (§4).

One key advantage of SutraNets is that training *decomposes* over sub-series. Sub-series RNNs do not condition on hidden states from other sub-series RNNs, rather only on *generated* values. During training, when we have access to the true values of other sub-series, all inputs are known in advance and sub-series RNNs can be trained in *parallel*. The only sequential action is evolution of each RNN's hidden state — over $1/K$ fewer steps than standard or multidimensional RNNs [78, 81]. In this way, SutraNet-RNNs provide middle-ground between RNNs and standard Transformers (with no sequential computation in training). Inference in SutraNets, like all AR models, is sequential [36].

**Application to Transformers** Rather than using $K$ RNNs to parameterize the conditional probabilities as in Eq. (3), we could instead use $K$ autoregressive Transformers. Compared to a standard Transformer, backfill and non-alternating sub-series Transformers would have the advantage of reducing *error accumulation* (by increasing the generative stride), but not of *signal path* (already $\mathcal{O}(1)$ for Transformers [82]). Moreover, at each timestep, a sub-series Transformer could attend to essentially all prior values, limited only by the generative order. In a naive implementation, SutraNet Transformers would therefore have an asymptotic complexity of $\mathcal{O}(N^2)$ — the same as standard Transformers. However, we can attain $\mathcal{O}(N^2/K)$ complexity by restricting each sub-series Transformer to only attend to values from its own sub-series, plus a few proximal values from other sub-series (similar to strided or banded self-attention [12, 8]). Although asymptotically larger than the $\mathcal{O}(N(\log N)^2)$ of LogSparse attention [44], it merits empirical investigation to determine which approach achieves the most favorable accuracy vs. complexity trade-off.

**Limitations and broader impact** SutraNets are more complex than standard sequence models in a few ways. First, they require more decisions, such as specifying the number of sub-series. While superior accuracy can be obtained when making these choices heuristically, we evaluate the extent to which tuning can achieve further gains (§4). Secondly, having $K$ separate RNNs results in roughly $K$ times as many total parameters. Although, as noted above, this does not translate into requiring more memory or training/inference time, SutraNet models are larger to store and transmit.

As a flexible predictor of both long and short horizons (and without needing *a priori* specification of confidence quantiles), SutraNets could also serve as the basis for a general-purpose large forecasting model (LFM), along the lines of LLMs like GPT3 [8]. Like modern LLMs, LFMs could potentially be used with little task-specific data and no fine-tuning, and may have similarly-large environmental and financial costs [4]. As with LLMs, new techniques will be needed to detect [64, 71, 54], document [51, 22], and mitigate [25, 58] unsafe usage and other failure cases.

## 4 Experiments

**Training.** We implemented SutraNets in PyTorch [59]. We use Adam [37] (default $\beta_1$=0.9, $\beta_2$=0.999, $\epsilon$=$10^{-8}$), early stopping, and a decaying learning rate ($\gamma$=0.99). The default RNN is a 1-layer LSTM with 64 hidden units; when using >1 layers (Table 3), inter-layer dropout [50] with $p_{drop}$=0.001 is applied. For every model/dataset pair, we tune weight decay and initial learning rate over a $4 \times 4$ grid.

**Systems.** We evaluate the following systems. Naïve and SeasNaïve are non-probabilistic, untrained baselines, while the C2FAR variants follow the standard RNN ordering (Fig. 2a):

| | |
|---:|:---|
| Naïve: | At each horizon, repeats last-observed historical value [34]. |
| SeasNaïve: | At each horizon, repeat previous value at same *season* as horizon [34], e.g., same hour-of-day (SeasNaïve1d) or hour-of-week (SeasNaïve1w). |
| C2FAR: | LSTM-based forecasting model [7], configured as described above (§3). |
| C2FAR+lags: | C2FAR, but with extra features for previous hour-of-day values (C2F-encoded). |
| C2FAR+dropout: | C2FAR, but with $p_{drop}$=0.2 (and re-scaling [74]) applied to inputs during training. |
| Low2HighFreq: | High-frequency forecast generated conditional on low-frequency values (§3). |
| SutraNets: | Variations Regular-alt, Regular-non, Backfill-alt, Backfill-non (§3). |

**Datasets.** We evaluate on the following datasets, where we note the number of SutraNet sub-series, $K$, as well as lengths of conditioning, $C$, and prediction, $P$, ranges:

| | |
|---:|:---|
| *azure*: | $K$=6, $C$=2016, $P$=288: 5-minutely usage of 2085 VM flavors, groupings (by tenant, etc.) in Azure cloud. Originally from [15], aggregated here instead to 5-minute intervals. |
| *elec*: | $K$=6, $C$=168, $P$=168: Hourly electricity usage of 321 customers [17], version from [39]. |
| *traff*: | $K$=6, $C$=168, $P$=168: Hourly occupancy for 862 car lanes [17], version from [89]. |
| *wiki*: | $K$=7, $C$=91, $P$=91: Daily hit-count for 9535 pages, first used in [21], version from [27]. |
| *mnist*: | $K$=4, $C$=392, $P$=392: row-wise sequential pixel values for 70,000 $28 \times 28$ digits [42]. |
| *mnist*$^\pi$: | $K$=4, $C$=392, $P$=392: *mnist* sequences, but with a fixed, random permutation applied. |

Table 2: ND%, wQL% across datasets, best results in **bold**. For *mnist/mnist$^\pi$*, previous *row* value used for SeasNaïve1d. Poor results for C2FAR relative to Regular-alt indicate *signal path problem* (green cells). Poor results for both C2FAR and Regular-alt indicate *discrepancy problem* (pink). Non-alternating weakness indicates *missing info problem* (yellow). Backfill-alt reduces signal path and discrepancy without missing info, therefore offering best or near-best solution on each dataset.

| | *azure* | *elec* | *traff* | *wiki* | *mnist* | *mnist$^\pi$* |
|---|---|---|---|---|---|---|
| Naïve | 3.2, 3.2 | 43.0, 43.0 | 75.5, 75.5 | 44.0, 44.0 | 100.0, 100.0 | 117.9, 117.9 |
| SeasNaïve1d | 2.8, 2.8 | 10.5, 10.5 | 31.1, 31.1 | 44.0, 44.0 | 145.2, 145.2 | 163.5, 163.5 |
| SeasNaïve1w | 5.2, 5.2 | 11.1, 11.1 | 17.5, 17.5 | 41.8, 41.8 | N/A | N/A |
| C2FAR | 3.2, 2.5 | 10.6, 8.4 | 19.3, 16.0 | 31.1, 27.2 | 67.9, 52.3 | 100.0, 95.7 |
| C2FAR+lags | 2.7, 2.2 | 10.4, 8.3 | 18.0, 15.0 | 31.2, 27.2 | 68.1, 52.8 | 99.3, 89.9 |
| C2FAR+dropout | 3.2, 2.6 | 10.9, 8.7 | 18.5, 15.3 | 31.3, 27.0 | 68.6, 53.3 | 99.7, 90.4 |
| Low2HighFreq | 2.6, 2.1 | 10.3, 8.2 | 20.2, 16.6 | 30.9, 27.2 | 62.4, 47.6 | 98.1, 88.3 |
| Regular-alt | 3.3, 2.6 | 9.9, 7.9 | 15.5, 13.0 | 30.4, 26.4 | 67.9, 52.4 | **48.7, 38.3** |
| Regular-non | 2.6, 2.1 | 9.7, 7.8 | 15.6, 13.1 | 31.2, 27.3 | 59.4, 45.4 | 90.1, 69.7 |
| Backfill-alt | 2.5, 2.0 | **9.3, 7.4** | **15.3, 12.8** | **30.1, 26.1** | 64.4, 49.7 | 72.0, 54.6 |
| Backfill-non | **2.3, 1.9** | **9.3, 7.4** | 15.7, 13.1 | 30.5, 26.8 | **58.9, 44.9** | 78.2, 58.3 |

Sequential (binarized) MNIST has been used for NLL [40] and classification [41] evaluations. Le et al. [41] used a fixed permutation "to make the task even harder." Here, we forecast a distribution over *real-valued* pixel intensities in the second half of an image, based on values in the first. Unlike the other datasets, train/dev/test splits are not based on time, rather we use the standard test digits [42].

**Metrics.** We evaluate the median of our forecast distribution by computing *normalized deviation* (ND) from true values. We evaluate the full distribution using *weighted quantile loss* (wQL), evaluated at forecast quantiles $\{0.1, 0.2 \ldots 0.9\}$, as in [60, 7]. For non-probabilistic Naïve and SeasNaïve baselines, note wQL reduces to ND, similarly to how CRPS reduces to absolute error [24, §4.2].

**Results.** We summarize the most important experimental findings as follows:

> *Finding 1: Backfill-alt and Backfill-non improve over C2FAR and other baselines on every dataset, with both obtaining an average relative reduction in ND of around 15% over the best non-SutraNet.*

Table 2 has the main results; note the differences in ND% are very stable: we repeated *grid search tuning and testing* multiple times with different random seeds and found *SutraNets superior to standard models across all repeats* (supplemental §B.2). Backfill-non does best when the training/inference discrepancy problem predominates (*azure* & *mnist*, Table 2). Adverse effects of discrepancy seem to be magnified by high data *redundancy*; in these cases, non-alternating generation reduces discrepancy while having sufficient info to forecast accurately. We also hypothesize that Regular-non performs poorer because it generates less-*coherent* output: by construction, it only conditions on past values, even when subsequent values have already been generated (e.g., row 7 in Fig. 2d). In contrast, when generating a given point (e.g., row 7 in Fig. 2e), Backfill-non has always seen the previous and future proximal values, which were input at the previous and current timesteps, respectively. When signal path problems predominate (*traff* & *mnist$^\pi$*), alternating generation, with access to more info, performs better. On *mnist$^\pi$*, differences between regular and backfill depend solely on the random permutation; the superiority of Regular-alt here motivates tuning over multiple orderings of sub-series, perhaps averaging over them, as is done in some prior non-sequential models [23, 80].

Accuracy gains seem to derive from solving both discrepancy *and* signal path problems together. When discrepancy predominates, Regular-alt performs poorly, even though signal path is improved. For the opposite case, where error accumulation alone is improved, we implemented an RNN that processes values in backfill order, but otherwise acts as a standard RNN. This model performs much poorer than SutraNet-based Backfill-alt (supplemental §B.4). The lack of a holistic solution may also explain the inconsistency of lags and dropout, which sometimes help, sometimes hurt compared to vanilla C2FAR. Low2HighFreq works best on datasests where discrepancy predominates, while performing much poorer when signal path is important. Note that weekly-seasonal baselines have not previously been evaluated on these datasets. Observing in particular the relatively strong accuracy of SeasNaïve1w on *traff*, we suggest they be included in future evaluations.

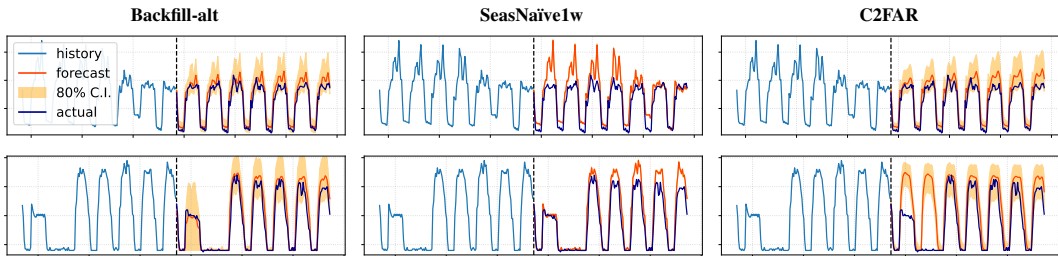

Figure 3: Forecasts and 80% intervals for *elec* series. Backfill-alt performs well with a changepoint (row 1, where SeasNaïve1w fails) and with weekly seasonality (row 2, where C2FAR fails).

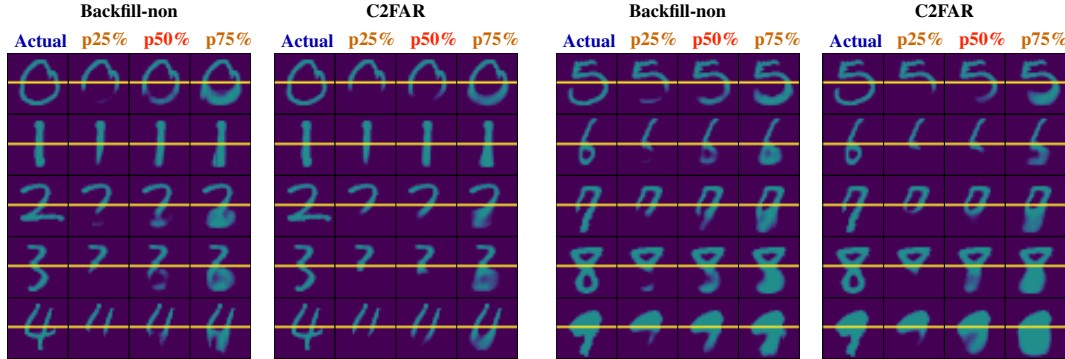

Figure 4: Percentiles of forecast distributions for first occurrence of 0 . . . 9 in *mnist* test data. Pixels before yellow line (in row-order) are history. Backfill-non more confident in bottom part of digits.

> **Finding 2**: *SutraNets are able to effectively model both long and short-term dependencies.*

On *elec* data, where SeasNaïve1d is more effective than longer-term SeasNaïve1w (Table 2), modeling long-term dependencies remains important (Fig. 3). Backfill-alt is able to capture both short-term changes and long-term seasonality, the latter of which is neglected by C2FAR. In plots of the *mnist* forecasts (Fig. 4), we see that compared to C2FAR, Backfill-non generates more non-zero values in the bottom parts of digits 0, 2, 5, etc., e.g., at p50%, implying the networks have more certainty or *confidence* (i.e., tighter prediction intervals) on the digit being generated. Also, Table 2 results indicate that discrepancy is the predominant problem on *mnist*; this may help explain why autoregressive models like PixelRNN [81] score very high in likelihood-based evaluation (where there is no sampling), but produce less-realistic samples compared to GANs. SutraNets may therefore provide a useful tool for reducing training/inference discrepancy in a variety of image generators.

> **Finding 3**: *SutraNets are more accurate than C2FAR across all forecast horizons.*

Autoregressive models are normally thought to accumulate and compound [56] errors "along the generated sequence" [5]. We therefore expect the accuracy gap between SutraNets and C2FAR to grow across horizons. Instead, on *elec* and *traff*, we already see a large gap over the first 24 hours (supplemental §B.3). Improvements in both informational asymmetry (§1) and signal path evidently enable gains across all steps. Low2HighFreq, in contrast, is only accurate at horizons corresponding to its low-freq forecast, while Backfill-alt is slightly better at horizons corresponding to its first sub-series. This suggests errors do "accumulate" to a small extent across sub-series autoregression.

> **Finding 4**: *Deeper models enable improved long-sequence forecasting, for SutraNets and C2FAR.*

In particular, going from 1 to 2 layers greatly improves accuracy on *traff* (Table 3), a dataset where signal path is the predominant problem. Smaller improvements are also seen on *elec*. Since both LSTM stacking and use of SutraNets increase complexity, a natural question is which approach offers the best computational tradeoff for a desired level of accuracy. Fig. 5 shows that for a given number of parameters — and especially for a given inference time — SutraNets are able to obtain better

Table 3: ND% decreases for SutraNets & C2FAR as number of RNN layers/hidden units increases.

| Dataset | *elec* | | | | *traff* | | | |
|---|---|---|---|---|---|---|---|---|
| #layers | 1 | 2 | 3 | 4 | 1 | 2 | 3 | 4 |
| #hidden | 64 | 128 | 256 | 256 | 64 | 128 | 256 | 256 |
| C2FAR | 10.6 | 10.2 | 9.9 | 9.7 | 19.3 | 14.7 | 14.8 | 14.4 |
| Regular-alt | 9.9 | 9.3 | **9.0** | 9.2 | 15.5 | 14.3 | 14.3 | 14.2 |
| Backfill-alt | 9.3 | 9.3 | **9.0** | 9.3 | 15.3 | 14.2 | 14.1 | **14.0** |
| Backfill-non | 9.3 | 9.1 | 9.5 | 9.3 | 15.7 | 14.4 | 14.4 | 14.4 |

Table 4: ND% lowest when #sub-series divides into seasonal period (24), but alternating models less sensitive on *elec* (cf. C2FAR with 10.6% on *elec*, 19.3% on *traff*.)

| Dataset | *elec* | | | *traff* | | |
|---|---|---|---|---|---|---|
| #subseries | 6 | 7 | 12 | 6 | 7 | 12 |
| Regular-alt | 9.9 | 10.0 | 10.0 | 15.5 | 17.2 | 15.0 |
| Backfill-alt | **9.3** | 9.4 | 9.4 | 15.3 | 16.9 | **14.9** |
| Backfill-non | **9.3** | 10.0 | 9.5 | 15.7 | 17.3 | 15.4 |

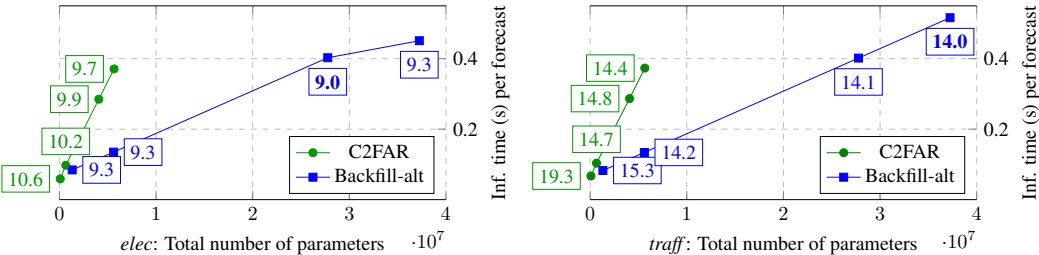

Figure 5: Total parameters (x-axis) and average inference time per forecast (y-axis, lower better) for C2FAR and Backfill-alt systems of Table 3, for *elec* (left) and *traff* (right). ND% labeled at each point in boxes. Backfill-alt enables more parameters and better ND% at equivalent speeds.

ND% (plots of memory usage and of training time are similar, see supplemental). SutraNet speed and memory usage scale with the complexity of each RNN rather than the number of sub-series RNNs (§3); they can consequently make use of $K\times$ more parameters without $K\times$ the cost.

> ***Finding 5***: *Backfill-alt is robust across different configurations, including when we vary $K$.*

We also experimented with varying the number of sub-series. For all $K>1$, SutraNets work much better than standard C2FAR (Table 4). Interestingly, $K=12$ led to the highest accuracy on *traff*, where the signal path problem predominates. When $K$ does not divide evenly into the seasonal period, non-alternating models have a missing info problem (e.g., they miss predictive preceding values at the same hour-of-day). Both Backfill-alt and Regular-alt can access these values through covariate features (cf. Fig. 2f), and hence perform relatively better when $K=7$ on *elec*. Backfill-alt is relatively weaker at $K=7$ when there is less redundancy (*traff*), although still superior to standard C2FAR.

See the supplemental for further experimental details and results. Code for SutraNets is available at `https://github.com/huaweicloud/c2far_forecasting/wiki/SutraNets`.

## 5  Conclusion

Probabilistic forecasting of long time series is an under-studied problem, with most recent work focusing on point prediction. We presented SutraNets, a novel autoregressive approach to probabilistic forecasting of long-sequence time series. SutraNets convert a univariate series into a $K$-dimensional multivariate series, each dimension comprising a distinct every-$K$th-value sub-series of the original sequence. The SutraNet autoregressive model generates each sub-series conditional on both its own prior values, and on other sub-series, ensuring coherent output samples. From these samples, an estimate of the full joint probability of future values is made. SutraNets leverage C2FAR under-the-hood, allowing for efficient representation of time series amplitudes, and low-overhead input encoding of covariate sub-series. SutraNets provide a holistic solution to challenges in long-sequence sampling, reducing training/inference discrepancy (by generating conditional on longer-term information) and signal path distances (by a factor of $K$). Experimentally, SutraNets achieve state-of-the-art results on a variety of long-sequence forecasting data sets, while demonstrating similar speed and memory requirements as standard sequence models, even when using $K\times$ more parameters.

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
