# SutraNets: Sub-series Autoregressive Networks for Long-Sequence, Probabilistic Forecasting: Supplementary Materials

**Shane Bergsma**    **Timothy Zeyl**    **Lei Guo**
Huawei Cloud, Alkaid Lab Canada
{shane.bergsma,timothy.zeyl,leiguo}@huawei.com

## A   Experimental details

### A.1   Architecture

As mentioned in §3 of the main paper, we use C2FAR-LSTMs [2] for our sub-series RNNs. The detailed architecture of the LSTMs follows the description in [2, supplement]. In particular, network layers include *bias weights*, and the same number of hidden units are used in each LSTM layer when multi-layer LSTMs are used. Like C2FAR, we also follow DeepAR [11] in using the same network to encode (i.e., process the conditioning range) and decode (i.e., generate values in the prediction range). Like C2FAR, during training we only compute loss over the prediction range.

### A.2   Form of output distribution and input encoding

As mentioned in §3 of the main paper, when training DeepAR-style models, values in the conditioning and prediction ranges are normalized based on the amplitudes in the conditioning range. Likewise, during inference, conditioning values are normalized based on the conditioning range; forecasts are subsequently made in the normalized space before they are ultimately unnormalized in order to create the final output sample. In SutraNets, we use min-max scaling [8] to normalize values for a target sub-series, based on the min and max of the conditioning range *of that sub-series*. In other words, each sub-series is forecast in its own normalized space. Theoretically, this could be advantageous if the sub-series have very different amplitudes as it would allow each sub-series to make use of the full range of bins in the coarse-to-fine discretization, ultimately increasing the precision of the forecasts.

However, also recall that for each sub-series RNN, at each step we encode and provide as inputs both previous values of that sub-series (the *target* sub-series), as well as covariate features from *other* sub-series (autoregressively). We therefore have two options for normalizing the covariate values from the other sub-series: (1) normalize these values according to the dynamic range in the conditioning range of the *target* sub-series, or (2) normalize these values according to the dynamic range of *their own* conditioning range. In preliminary experiments on validation data, we found the former approach to be slightly more effective, so adopt this approach with SutraNets. The advantage of target-specific normalization is that covariate sub-series values are always normalized *consistently* with the target sub-series. The disadvantage is that any covariate sub-series that has very different amplitudes than the target could become normalized to very high or very low values; this could result in the discretization of all such values to only a few bins, and therefore only very coarse information from the covariate series would be conveyed by the covariate features. In future work, we plan to investigate this issue further and ascertain whether other normalization strategies could prove more effective in certain cases.

### A.3   Training

37th Conference on Neural Information Processing Systems (NeurIPS 2023).

Table 1: Fixed hyperparameters.

| Hyperparameter | Value | Note |
|---|---|---|
| n_train_batch_size | 128 | Total num. prediction ranges per training batch |
| n_train_ranges_per_checkpoint | 8192 | Total num. prediction ranges in one *checkpoint* (train loss reported) |
| n_max_checkpoints | 750 | Maximum num. checkpoints (n_train_ranges_per_checkpoint sets) |
| n_rollouts (validation) | 25 | Num. samples for sampling when evaluating on validation set |
| n_validation_set | 8192 | Total num. prediction ranges per validation evaluation |
| n_stop_evals_no_improve | 37 | Num. validation evals without improvement before early stop |
| n_rollouts (test) | 500 | Num. samples for sampling when evaluating on test set |

Table 2: Tuning ranges for hyperparameter grid-search optimization.

| Hyperparameter | Range |
|---|---|
| learning_rate | [1e-4, 1e-3, 1e-2, 1e-1] |
| weight_decay | [1e-7, 1e-6, 1e-5, 1e-4] |

### A.3.1 Slicing of training windows

As mentioned in §3 of the main paper, recall that SutraNets, like other autoregressive forecasting models, are trained by slicing many training series into many *windows*, i.e., conditioning+prediction ranges at different start points. During training, we randomly select windows for training batches *without replacement*, until all such windows have been exhausted, at which point we repeat the random slicing process.

These windows are clock-aligned in the sense that, for example, the $i$th hourly value in a window is assumed to correspond to some hour of the day (e.g., 2pm-3pm). However, as noted in Footnote 2 of the main paper, these windows do not begin and end at *fixed* hours of the day; one window may begin at 2pm (and span two weeks) and the next may begin at 7am (and again span two weeks). This means each sub-series sequence model does not learn hour-of-day-specific (or, more generally, season-specific) patterns.

### A.4 Training parallelism

We vectorize across multiple conditioning+prediction windows during training. The number of windows that we parallelize over is referred to as the *n_train_batch_size* (currently set to 128, see Table 1). We evaluate 8192 windows during each training checkpoint (*n_train_ranges_per_checkpoint*), and train for a maximum of 750 checkpoints (*n_max_checkpoints*).

### A.5 Tuning

As mentioned in §4 of the main paper, we tune the hyperparameters weight decay and initial learning rate over a $4{\times}4$ grid. Tuning via grid search is commonly performed in forecasting [13, 14, 9, 10]. The specific values used in our grid are given in Table 2. We tune directly for normalized deviation (ND) on validation data, evaluating *after every training checkpoint*. ND evaluation requires running the Monte Carlo sampling procedure in order to generate a forecast distribution (§A.9); we use the median of this forecast distribution as the point forecast for evaluation. We use *n_rollouts*=25 samples in the Monte Carlo estimate, over a fixed validation set of *n_validation_set*=8192 prediction ranges. We stop a tuning trial early if we see *n_stop_evals_no_improve* evaluations without a new top score (currently set to 37, see Table 1). As training times are roughly comparable for C2FAR and SutraNets (Fig. 2), total tuning cost is similar for both approaches, as well as for C2FAR+lags, C2FAR+dropout, and Low2HighFreq.

### A.6 Evaluation

We use 500 separate rollouts during the forecasting process on held-out data (*n_rollouts*=500). We compute rolling evaluations with a stride of 1, i.e., we forecast and evaluate over overlapping prediction ranges, as in [5].

Table 3: Dataset details. *Note: while other datasets use disjoint temporal periods for Dev and Test data, totally disjoint *series* (MNIST images) are used for the Dev and Test sets of *mnist* and *mnist$^\pi$*.

| Dataset | Domain | Freq | Num. series | Vals per series | Dev vals per series | Test vals per series | Condit. range size | Pred. range size |
|---------|--------|------|-------------|-----------------|---------------------|----------------------|--------------------|-------------------|
| *elec* | Discrete | Hourly | 321 | 21212 | 504 | 504 | 168 | 168 |
| *traff* | Real | Hourly | 862 | 14204 | 504 | 504 | 168 | 168 |
| *wiki* | Discrete | Daily | 9535 | 912 | 91 | 182 | 91 | 91 |
| *azure* | Discrete | 5-minute | 4048 | 8628 | 864 | 864 | 2016 | 288 |
| *mnist* | Real | 1 pixel | 70000 | 784 | 784* | 784* | 392 | 392 |
| *mnist$^\pi$* | Real | 1 pixel | 70000 | 784 | 784* | 784* | 392 | 392 |

Table 4: Dataset-specific parameter settings for different systems.

| Dataset | C2FAR Binning Low | C2FAR Binning High | SutraNets Binning Low | SutraNets Binning High | Lag Period |
|---------|-------------------|--------------------|-----------------------|------------------------|------------|
| *elec* | -0.01 | 1.06 | -0.06 | 1.20 | 24 (one day) |
| *traff* | -0.01 | 1.01 | -0.02 | 1.23 | 24 (one day) |
| *wiki* | -0.16 | 2.34 | -0.79 | 5.13 | 7 (one week) |
| *azure* | -0.05 | 1.15 | -0.08 | 1.20 | 288 (one day) |
| *mnist* | 0.00 | 1.00 | 0.00 | 1.00 | 28 (one row) |
| *mnist$^\pi$* | 0.00 | 1.00 | 0.00 | 1.00 | 28 (one row) |

## A.7  Computational resources

SutraNets are implemented in PyTorch [7], version `1.9.1+cu102`. We use GPUs from Nvidia: four Tesla P100 GPUs with 16280MiB and two Tesla K80 GPUs with 11441MiB.

## A.8  Datasets

### A.8.1  Azure VM demand dataset

The *azure* dataset was first used for forecasting in [2]; this work leveraged the publicly-available *Azure Public Dataset*[1], originally released in [4] under a Creative Commons Attribution 4.0 International Public License. We converted the event stream in the Azure Public Dataset into time series by exactly following the approach in [2, supplement], except rather than aggregating the data over a 1-hour period, we aggregated the data over a 5-minute period. This is actually the most precise aggregation possible given the original dataset quantizes all timestamps using 5-minute precision. We also used the same experimental splits as in [2], using 20 days as training, 3 days for validation, and 3 final days for testing. Compared to hourly granularity, with 5-minute intervals, there are $12\times$ as many rolling windows to evaluate, and each has $12\times$ as many steps. To alleviate the computational burden for this dataset, we evaluate all systems on a random, fixed 53% subset of the 4.1M+ windows in the test period.

### A.8.2  Other datasets

The *elec*, *traff*, and *wiki* datasets were obtained using scripts in GluonTS [1]. Compared to the training/validation/test splits used in prior work [10, 8, 5], here we use more validation/test values (Table 3), reflecting the longer forecast horizons that we evaluate on.

The *mnist* dataset was obtained from [6]. The standard 10,000 test images were used as our test set, while a random 10,000-element subset of the 60,000 training images were used as a validation set. The *mnist$^\pi$* dataset was obtained by applying a fixed random permutation to every element of these same datasets. The pixel values were used in their original floating-point format, i.e., they were not binarized or modified in any way beyond the `ToTensor()` transform in `torchvision`. Also, note

---

[1]`https://github.com/Azure/AzurePublicDataset/blob/master/AzurePublicDatasetV1.md`

that even though *mnist* and *traff* dynamic ranges are actually bounded (between 0 and 1), we do not use this information explicitly; that is, we read the conditioning ranges and normalize in the same way as we do on all datasets.

Table 3 provides the details of these datasets and *azure*. In Table 4, we note some system-specific configurations for each dataset. To help explain this table, we now provide some background information. First of all, note that a C2FAR binning always has a fixed extent from a low to a high cutoff, over the min-max normalized values [2]. The binning extent is selected in order to cover from roughly the 1% to the 99% percentiles of normalized values in the conditioning and prediction ranges of training data for each series. These values are normalized, as noted above, using min-max values from the *conditioning* range; the prediction range can go below the min and above the max. Also as noted above, recall that in SutraNets, each sub-series is normalized using the conditioning min and max values *of that sub-series*. We see in Table 4 that this generally corresponds to a wider binning range than with standard C2FAR, likely because there is greater variance between conditioning and prediction ranges when using the shorter sub-series sequences. Table 4 also gives the lag period used in C2FAR+lags.

## A.9   Metrics

Let $y_{i,t}$ be the $t$th value of the $i$th time series, that is, $i$ indexes over time series and $t$ indexes over time steps. Recall that autoregressive forecasting approaches create a Monte Carlo estimate of the forecast distribution by repeatedly sequentially sampling the model in the prediction range. Forecast quantiles at a given horizon are then estimated by calculating quantiles of the sampled forecasts at that horizon. Let $\alpha$ represent the quantile of interest, e.g., $\alpha$=0.5 indicates we want quantile 0.5, i.e., the 50th percentile, while $\alpha$=0.9 indicates the 90th percentile, etc. Let $\hat{y}_{i,t}^{(q_\alpha)}$ be the actual estimated $\alpha$ quantile of the forecast distribution for time series $i$ at point $t$, e.g. $\hat{y}_{i,t}^{(q_{0.9})}$ is the value such that 90% of possible values for point $y_{i,t}$ are expected to be below this value (and, as noted above, we obtain this estimate from quantiles of our Monte Carlo samples). Our evaluation metrics always involve comparing an estimated quantile of the forecast distribution, $\hat{y}_{i,t}^{(q_\alpha)}$, at some horizon of the forecast, to an observed true value at that horizon $y_{i,t}$. For normalized deviation (ND), we compare the 50th percentile of the forecast distribution to the observed true value, i.e., we use the 50th percentile as a *point estimate*. For wQL, following [8, 2], we compare multiple estimated quantiles (at $\alpha=\{0.1, 0.2 \ldots 0.9\}$), to the same observed true value.

More formally, let $\mathcal{I}(\cdot)$ denote the indicator function. We define *pinball loss* and *quantile loss* as part of the derivation of *weighted quantile loss*. *Weighted quantile loss* and *normalized deviation* are reported in the main paper, as percentages.

**Pinball loss:**
$$\Lambda_\alpha(\hat{y}_{i,t}^{(q_\alpha)}, y_{i,t}) = (\alpha - \mathcal{I}(y_{i,t} < \hat{y}_{i,t}^{(q_\alpha)}))(y_{i,t} - \hat{y}_{i,t}^{(q_\alpha)})$$

**Quantile loss:**
$$\mathrm{QL}_\alpha = \frac{\sum_{i,t} 2\Lambda_\alpha(\hat{y}_{i,t}^{(q_\alpha)}, y_{i,t})}{\sum_{i,t} |y_{i,t}|}$$

**Weighted quantile loss:**
$$\mathrm{wQL} = \frac{1}{9}(\mathrm{QL}_{0.1} + \mathrm{QL}_{0.2} + \cdots + \mathrm{QL}_{0.9})$$

**Normalized deviation:**
$$\mathrm{ND} = \frac{\sum_{i,t} |y_{i,t} - \hat{y}_{i,t}^{(q_{0.5})}|}{\sum_{i,t} |y_{i,t}|}$$

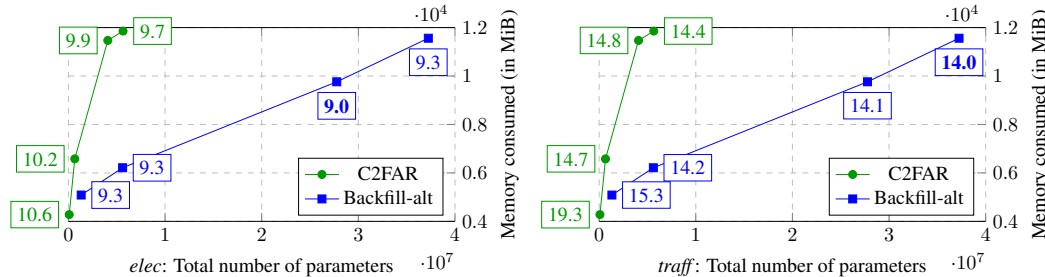

Figure 1: Total number of parameters (x-axis) and total *memory consumed* during inference (y-axis, lower is better), measured via `nvidia-smi` on NVIDIA Tesla P100, in MiB, for C2FAR and Backfill-alt systems of Table 3 in the main paper, for *elec* (left) and *traff* (right). Accuracy labeled at each point in ⌐boxes⌐, with best result in **bold**.

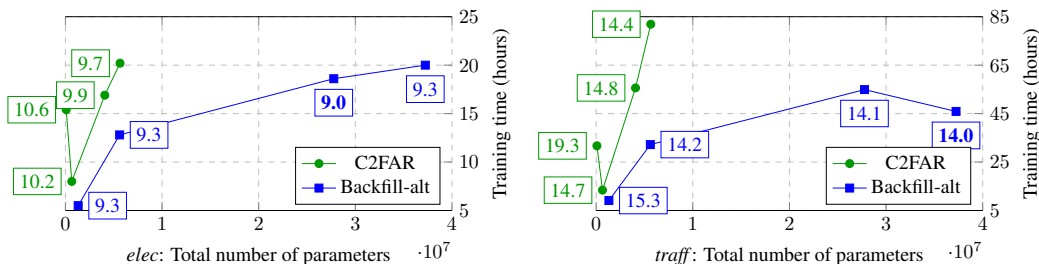

Figure 2: Total number of parameters (x-axis) and *training time* in hours (y-axis, lower is better) for C2FAR and Backfill-alt systems of Table 3 in the main paper, for *elec* (left) and *traff* (right). Accuracy labeled at each point in ⌐boxes⌐, with best result in **bold**.

# B Experimental results

## B.1 Computational performance and resource requirements

See Fig. 5 in the main paper for inference *time*, and Fig. 1 here for inference *memory usage*, for the systems of Table 3 in the main paper. All measurements of speed and memory were made on NVIDIA Tesla P100 GPUs, with a common test batch size of 32, and 500 Monte Carlo samples for the forecast distribution estimation.

Fig. 2 has the training times for the systems of Table 3 in the main paper. Training time naturally reflects both the speed of convergence in learning (number of training epochs) and the speed of operating the specific architecture.

Table 5: Number of parameters for C2FAR and different SutraNet variations, as the number of layers and hidden units in the LSTMs vary. Note the number of SutraNet parameters is not affected by the sub-series ordering (backfill versus regular).

| nlayer | nhidden | C2FAR | 6-alt | 6-non-alt | 7-alt | 7-non-alt | 12-alt | 12-non-alt |
|---|---|---|---|---|---|---|---|---|
| 1 | 64 | 81,830 | 1,320,420 | 905,700 | 1,734,026 | 1,153,418 | 4,631,496 | 2,806,728 |
| 1 | 128 | 261,926 | 3,230,436 | 2,400,996 | 4,155,914 | 2,994,698 | 10,442,184 | 6,792,648 |
| 1 | 256 | 917,030 | 8,819,940 | 7,161,060 | 11,064,074 | 8,741,642 | 25,602,504 | 18,303,432 |
| 2 | 64 | 181,670 | 1,919,460 | 1,504,740 | 2,432,906 | 1,852,298 | 5,829,576 | 4,004,808 |
| 2 | 128 | 658,214 | 5,608,164 | 4,778,724 | 6,929,930 | 5,768,714 | 15,197,640 | 11,548,104 |
| 2 | 256 | 2,496,038 | 18,293,988 | 16,635,108 | 22,117,130 | 19,794,698 | 44,550,600 | 37,251,528 |
| 3 | 64 | 281,510 | 2,518,500 | 2,103,780 | 3,131,786 | 2,551,178 | 7,027,656 | 5,202,888 |
| 3 | 128 | 1,054,502 | 7,985,892 | 7,156,452 | 9,703,946 | 8,542,730 | 19,953,096 | 16,303,560 |
| 3 | 256 | 4,075,046 | 27,768,036 | 26,109,156 | 33,170,186 | 30,847,754 | 63,498,696 | 56,199,624 |
| 4 | 64 | 381,350 | 3,117,540 | 2,702,820 | 3,830,666 | 3,250,058 | 8,225,736 | 6,400,968 |
| 4 | 128 | 1,450,790 | 10,363,620 | 9,534,180 | 12,477,962 | 11,316,746 | 24,708,552 | 21,059,016 |
| 4 | 256 | 5,654,054 | 37,242,084 | 35,583,204 | 44,223,242 | 41,900,810 | 82,446,792 | 75,147,720 |

Exact numbers of parameters for the different models (and others that have not been evaluated), are given in Table 5.

Looking holistically at all of the performance figures, we can conclude two things:

1. For the same inference time, memory usage, or training time, SutraNets are typically more accurate than standard C2FAR (comparing along horizontal lines of each plot).

2. For the same number of parameters, SutraNets are typically more accurate than standard C2FAR (comparing along vertical lines of *any* plot).

The only exception to this rule is the two-level C2FAR model on *traff* (scoring 14.7%), which is superior to the 1-level SutraNet model (15.3%) while using fewer parameters. This model is also arguably competitive with SutraNets in terms of training and inference time (but uses more memory than a more accurate SutraNet model at 14.2%). As noted in the main paper (§4), the signal path problem predominates on the *traff* dataset. For this problem, it seems depth and SutraNets both offer effective strategies for improving accuracy with minor increases in computational overhead. With depth and SutraNets together, the 2-level SutraNet already dominates the deepest vanilla C2FAR model along all performance dimensions, while using fewer parameters.

It is also worth noting here that the Regular-alt is a unique SutraNet in that the same sequence model parameters could theoretically be used for each sub-series model. That is, each sub-series LSTM could continue to predict a distinct every-$K$th-value of the full sequence, and each such LSTM could continue to evolve its own distinct hidden state, while conditioning on its own unique covariates. However, each of these LSTMs could use the same trained LSTM *parameters*. This is a consequence of the unique ordering of Regular-alt (and the fact that the starting points of conditioning+prediction windows are not clock-aligned in training, as noted in §A.3.1). As such, a Regular-alt LSTM that shares model parameters among its different sub-series models could use only $1/K$ of the parameters compared to other SutraNets.

## B.2 Stability of empirical results

In this section, we investigate the stability of our empirical results. Random seeds are used in both our training/tuning process (via random sampling of windows for training batches, §A.3.1) and our testing process (via Monte Carlo sampling of predicted future values). It is important to quantify the stability of these sources of randomness separately [3]. Regarding our training/tuning process, ideally, for each system on each dataset, we would repeat our entire grid search tuning procedure multiple times with different random seeds, allowing us to determine the end-to-end stability of our approach to model fitting. While such repetition is not practical to perform over all datasets and over all depth/subseries variations, given the total time required, we elected to perform this procedure on *elec* and *traff*, in order to get definitive quantification of stability on these two datasets, and through these findings obtain a sense of the overall stability of our experimental results.

Fig. 3 provides the tuning stability results. Backfill-alt is remarkably stable across tuning runs on both *elec* and *traff*, while C2FAR shows much greater variation. In general, we find our tuning results to be very stable: SutraNets are superior to C2FAR across all repeats.

Meanwhile, Fig. 4 provides the testing stability results. In evaluation, both Backfill-alt and C2FAR are extremely stable across different random seeds.

## B.3 Evaluation by forecast horizon

Fig. 5 shows the forecast error of the systems as a function of the forecast horizon, for all datasets. These plots provide an interesting perspective on the training/inference discrepancy versus signal path problems. On datasets where discrepancy predominates (*azure* and especially *mnist*, main Table 2), differences between C2FAR and SutraNets do seem to start small and grow over time. Meanwhile, when signal path problems predominate (*traff* and especially *mnist$^\pi$*), differences between C2FAR and SutraNets are immediately large. These observations are further evidence that SutraNets provide both a useful diagnostic for errors in long-sequence generation, and a useful solution to these errors.

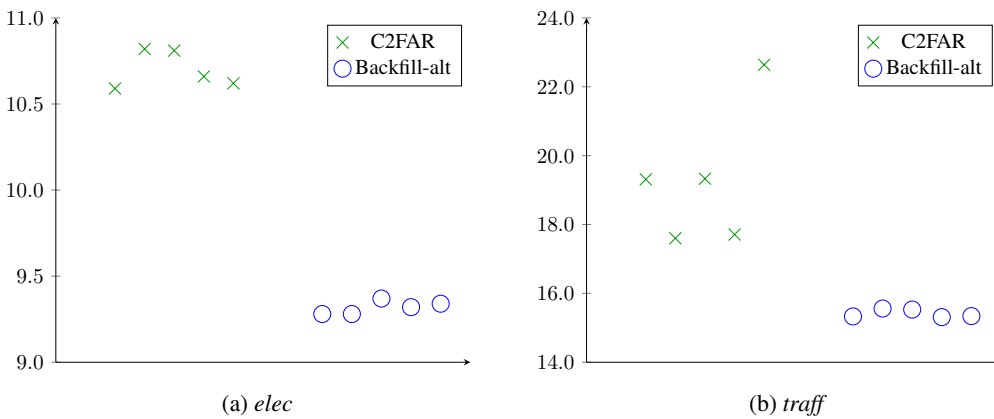

(a) *elec*    (b) *traff*

Figure 3: Tuning stability: ND% for the original *tuning* run (first marker) and four subsequent repeats with different random seeds, for both C2FAR and Backfill-alt, on *elec* (left) and *traff* (right). SutraNets have less variation across seeds than C2FAR.

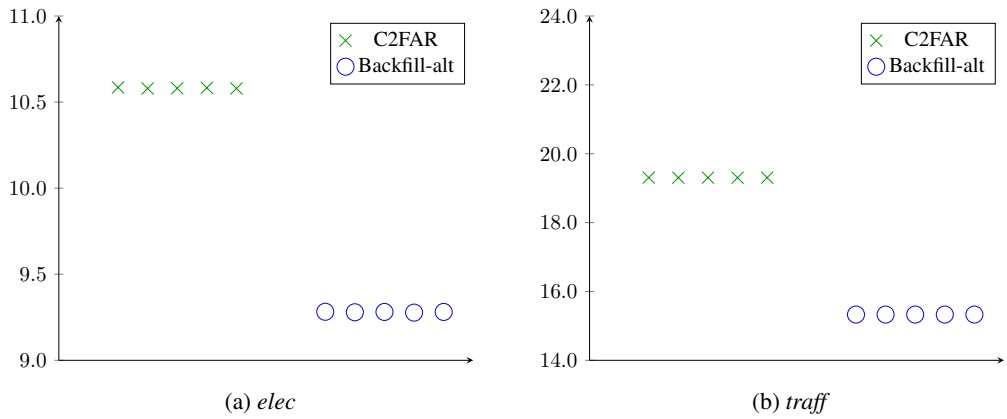

(a) *elec*    (b) *traff*

Figure 4: Testing stability: ND% for the original *evaluation* run (first marker) and four subsequent repeats with different random seeds, for both C2FAR and Backfill-alt, on *elec* (left) and *traff* (right). The variation across seeds is difficult to detect visually in all cases.

## B.4  Evaluation of a Backfill-Standard Model

Experimental results in the main paper (§4) clearly demonstrate improvements in forecasting accuracy when SutraNets are applied in backfill order. For example, Backfill-alt improves over Regular-alt on each of *azure*, *elec*, *traff* and *mnist* datasets. This raises an interesting question: could backfill ordering alone — i.e., used without SutraNets — lead to improvements over standard RNNs that process the values in regular order?

To investigate this question, we implemented a normal RNN model, with a single set of parameters and a single evolving RNN hidden state, but where we step through the time series in backfill order in segments of $K$ consecutive values. That is, within blocks of $K$ values, we visit the values in reverse order, and then move to the next block, akin to reading a document downwards but from right-to-left on each line. The resulting state transitions and feature dependencies are pictured in Fig. 6. This is essentially the same generative order as the Backfill-alt model in Fig. 2f of the main paper, but where a single state is updated at every value. We call this the Backfill-standard model.

Comparing the diagrams of Backfill-standard and Backfill-alt, it is clear that signal path will not be improved by Backfill-standard. However, because they both take $K$-step maximum *generative strides*, both approaches may have similar improvements in error accumulation. We therefore hypothesize that Backfill-standard may improve accuracy over the standard RNN, but remain less accurate than the Backfill-alt SutraNet. Such an improvement would be of significant practical importance, since

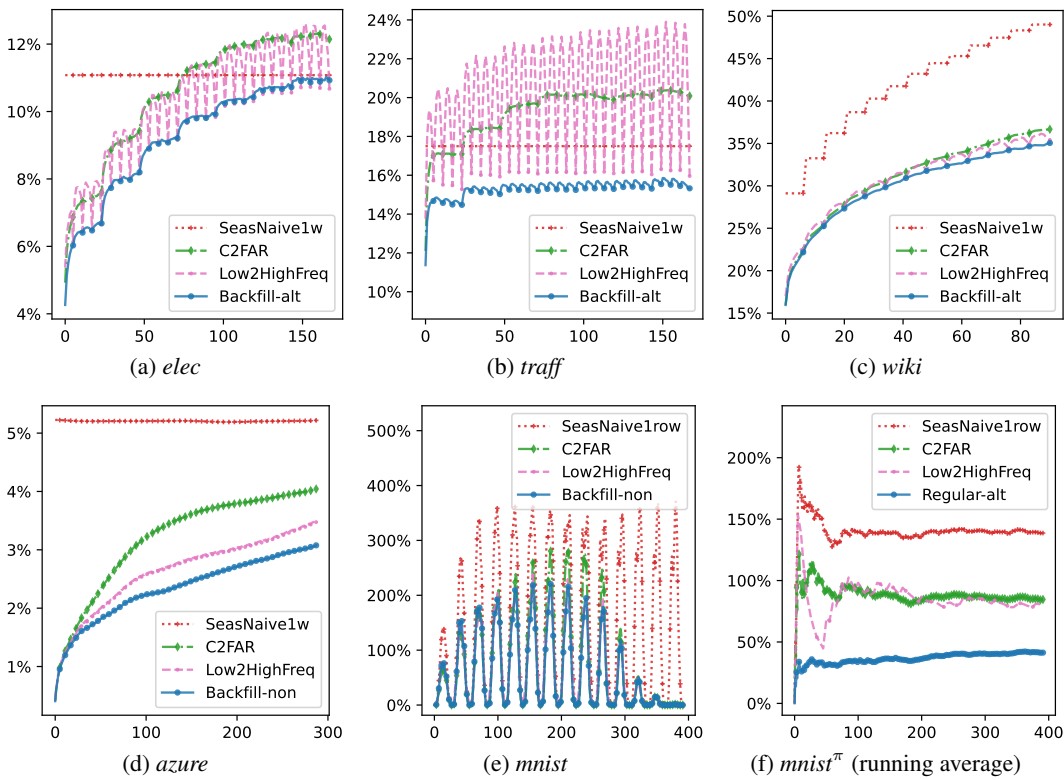

Figure 5: Normalized deviation (ND%) at different forecast horizons for all the datasets. Note the very cyclical nature on *mnist*, with a period of 28 — i.e., one row of the image — is due to the images being more predictable on the left/right edges of each row (where they are usually zero), but harder toward the center (where they are usually non-zero). Note also that we use the running average in *mnist*$^\pi$ since the original errors fluctuate very wildly by horizon.

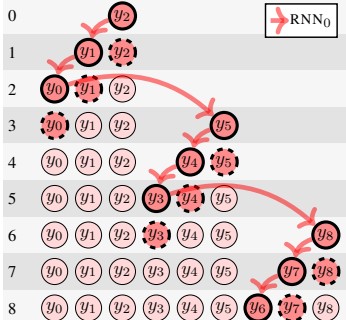

Figure 6: Ordering of generative steps in the Backfill-standard model, following same visual representations as Figure 2 in the main paper. That is, one output value (bold border) is generated in each row, conditional on (1) feature nodes (dashed border) *in that row*, and on (2) state from previous steps ($\Longrightarrow$ arrow connections). The Backfill-standard model does not use SutraNets; instead, it flips blocks of $K$ consecutive values into reverse (backfill) order, and then processes the values using a standard RNN.

Table 6: ND%, wQL% across three datasets for different RNNs. The Backfill-standard model performs worse than vanilla C2FAR and worse than all SutraNet variations across all datasets.

|  | *elec* | *traff* | *mnist* |
|---|---|---|---|
| C2FAR | 10.6, 8.4 | 19.3, 16.0 | 67.9, 52.3 |
| Regular-alt | 9.9, 7.9 | 15.5, 13.0 | 67.9, 52.4 |
| Regular-non | 9.7, 7.8 | 15.6, 13.1 | 59.4, 45.4 |
| Backfill-alt | 9.3, 7.4 | 15.3, 12.8 | 64.4, 49.7 |
| Backfill-non | 9.3, 7.4 | 15.7, 13.1 | 58.9, 44.9 |
| **Backfill-standard** | 11.0, 8.8 | 22.4, 18.1 | 83.2, 67.3 |

Table 7: ND%, wQL% across three datasets, *best* results in **bold**. Applying SutraNets via Backfill-alt improves both DeepAR-binned and C2FAR. C2FAR also improves over DeepAR-binned, and works synergistically with SutraNets.

|  | *elec* | *traff* | *mnist*$^\pi$ |
|---|---|---|---|
| DeepAR-binned | 10.6, 8.5 | 20.7, 17.1 | 82.8, 67.3 |
| DeepAR-binned+Backfill-alt | 9.6, 7.7 | 15.9, 13.3 | 76.4, 59.1 |
| C2FAR | 10.6, 8.4 | 19.3, 16.0 | 67.9, 52.3 |
| C2FAR+Backfill-alt | **9.3, 7.4** | **15.3, 12.8** | **64.4, 49.7** |

Backfill-standard is very straightforward to implement, simply requiring a kind of shuffling of the values in the conditioning and generation windows (i.e., a simple pre-processing step), which could be applied before using any standard sequence model.

To evaluate this question, we trained/tuned and tested Backfill-standard on three datasets, following the same experimental setup as we used for the main paper evaluations. Unfortunately, Backfill-standard performed quite poorly, worse than all SutraNets and worse than the standard C2FAR model (Table 6). One key point about Backfill-standard is that it not only does not *improve* signal path, but actually hurts it in many cases. For example, when generating value $y_3$ in row 5 of Fig. 6, the key information about the value of $y_2$ was provided many steps in the past (back in row 1, on the order of $2K$ steps in the past). The standard RNN, meanwhile, is always provided the $(i-1)$ input directly when generating the $i$th output, while Backfill-alt has access to $y_2$ at only one step in the past (as an input for the value generated at the previous step, see Fig. 2f in the main paper). Results for standard C2FAR and Backfill-standard are closer on *elec*, where signal path is less of an issue, but C2FAR still performs better. It seems the potential benefits in reducing error accumulation cannot be realized without also improving signal path; if the model cannot rely on longer-range information, it struggles to take accurate, longer predictive strides. These results provide further evidence that SutraNets are a useful approach because they solve both error accumulation and signal path together, and their benefits cannot be achieved with simple preprocessing tricks.

## B.5  Experiments with DeepAR-binned

Experimental results in the main paper (§4) show that SutraNets provide strong gains over the standard C2FAR model. In this section, we investigate whether a different distributional estimator, DeepAR-binned [8], also sees gains when used with SutraNets. To evaluate this question, we trained/tuned and tested both DeepAR-binned alone, and DeepAR-binned+Backfill-alt SutraNets. In both cases we use 1024 bins. We use the same three datasets as in §B.4 and again follow the same experimental setup as was used in the main paper evaluations.

Results are in Table 7. First of all, note that we replicate the findings of [2], but in the long-sequence setting: C2FAR alone performs better than DeepAR-binned alone in all cases (although ND% is quite close on *elec*). Secondly, we observe that applying SutraNets via the Backfill-alt ordering substantially improves DeepAR-binned on all datasets, demonstrating the broad applicability of the SutraNet approach. Finally, we note that SutraNets and the C2FAR enhancement are synergistic: applying both together results in the most accurate forecasting models. We also note that DeepAR-binned requires significantly more memory than the C2FAR system; DeepAR-binned uses 1024 output bins, while C2FAR uses only $12{\times}3$ output bins total via its efficient hierarchical factorization. Note the main computational bottleneck in tuning is sampling the output rollouts in order to compute ND%

Table 8: Comparison of CRPS for Informer and Scaleformer and wQL (a CRPS approximation) for other systems, using results from our main Table 2 and from Table 9 in Shabani et al. [12]. Since the implemented Scaleformer and Informer predictors cannot see values from one week (168 hours) ago, it makes sense that they perform much worse than both SutraNets, and the SeasNaïve1w baseline, on datasets with strong weekly seasonality. Practitioners and researchers should be aware of the potential cost of restricting the look-back context.

| System | Length of conditioning | Length of prediction | CRPS/wQL on *elec* | CRPS/wQL on *traff* |
|---|---|---|---|---|
| Informer [13] via [12, Table 9] | 96 | 96 | 0.330 | 0.372 |
| Scaleformer [12] via [12, Table 9] | 96 | 96 | 0.238 | 0.288 |
| SeasNaïve1w | 168 | 168 | 0.111 | 0.175 |
| Low2HighFreq (Scaleformer for RNNs) | 168 | 168 | 0.082 | 0.166 |
| SutraNets (Backfill-alt) | 168 | 168 | 0.074 | 0.128 |

on the validation set. Since DeepAR-binned requires more memory, we must perform this sampling over smaller batch sizes. The net result is DeepAR-binned tuning taking roughly 3X-4X more total GPU time to tune. C2FAR is therefore doubly synergistic with SutraNets: the combination allows for accurate, coherent forecasts, and also reduces the computational cost of these forecasts.

## B.6 Effects of shorter context windows

In the main paper, we mentioned that many recent systems for "long-term" forecasting use conditionining windows with a maximum length of 96 inputs, which amounts to *4 days* of history at 1-hour granularity. We explained that such restricted context is a serious limitation given that values from *7 days* in the past are highly predictive on datasets with strong weekly seasonality, as reflected in the good results for SeasNaïve1w on *elec* and *traff* in main Table 2.

Although Scaleformer with Transformers may be limited to short conditioning windows, the core idea of Scaleformer can be applied with RNNs; indeed, we implemented and evaluated a similar method to Scaleformer, but for RNNs (and using C2FAR as the output distribution), which we called Low2HighFreq, as noted in main paper §2. To illustrate the impact of decreased context windows, we compare the accuracy of Low2HighFreq and SutraNets in our work with those in Table 9 of Shabani et al. [12], where Informer and Scaleformer were used to generate *probabilistic* forecasts. We provide this comparison in Table 8.

Note that SutraNets predict 168 steps ahead vs. 96 for Scaleformer and Informer (which should disadvantage SutraNet results). Also note Shabani et al. [12] evaluate using CRPS, while we evaluate using wQL, a 10-point approximation to CRPS (for point predictors like seasonal-naive-1week, note both CRPS and wQL reduce to normalized deviation). It is problematic to compare these systems directly, as they have been trained and tuned in different ways, on different data splits, and with different output distributions. However, these results are nevertheless suggestive of the large potential drawbacks of using limited historical context. We hope that prior systems may find ways to use larger amounts of context and dramatically improve their forecasting accuracy.