# OpenReview forum: "SutraNets: Sub-series Autoregressive Networks for Long-Sequence, Probabilistic Forecasting"
_NeurIPS.cc/2023/Conference — NeurIPS 2023 poster_

### Official Review · Reviewer_BQ6D · 2023-07-03

**Soundness:** 3 good
**Presentation:** 4 excellent
**Contribution:** 3 good
**Rating:** 6
**Confidence:** 4

**Summary:**

The paper "SutraNets: Sub-series Autoregressive Networks for Long-Sequence, Probabilistic Forecasting" propose to model uni-variate time-series with two interleaved networks, that model 'fine-grained' and 'coarse-grained' time-steps. This has the advantage of reducing signal paths, reducing inference error accumulation, enable parallel training, and sharing parameters potentially more efficiently. The paper investigates 5 additional architectures on 6 standard datasets (e.g. electricity, but also MNIST), and demonstrates superior performance over the baselines.

**Strengths:**

+ the paper is well and very clearly written (including the literature review, methods, Figures etc.)
+ the core idea of the paper makes intuitive sense, and the paper proposes several good ideas how to extend the standard autoregressive architecture, and investigates their impact on the final performance.
+ the method is shown to improve performance over baselines

**Weaknesses:**

mainly minor things:
- Figure 4: it's a bit unclear what is meant with 'more confident' - I think it is probably meant that it is less blurry? I'd be great to mention this explicitly if this is the case. Also, it'd be great to somehow quantify these findings: what are the likelihoods on the hold-out test data in the different settings? Maybe in this case (MNIST), also image-based evaluations could be performed (e.g. FID).
- Figure 6: "per 100" in the figure itself. I think it'd be better to re-do the Figure with "inference time per forecast" and then put it there directly - otherwise it lacks a unit.
- There is some prior work on multi-variate forecasting that has some similarity to this paper, but hasn't been mentioned. E.g. [1] (please also check reference & citations of this work), though focussing on the full multi-variate time-series problem, do also use an 'inner' architecture to generate data, and then have an outer autoregressive model. These should be mentioned in the related literature.

[1] Rasul et al 2021. Multivariate Probabilistic Time Series Forecasting via Conditioned Normalizing Flows.

**Questions:**

- in eq.2 the conditioning is on x^k_{1:T+N}, i.e. y_{\tile t} is conditioned on co-variates from the future. I assume this should be x^k_{1:\tile t} instead?


**Limitations:**

+ The paper has a good limitations section, which covers the limitations. E.g. the method being more complex, and requiring additional tuning for better performance. Also the societal impacts are considered in this section.

---

> ### Author Rebuttal · Authors · 2023-08-09
>
> Thank you very much for your helpful and thoughtful review, and your positive comments regarding the paper's core idea, clarity of presentation, and thorough evaluation.
>
> ### Regarding concept of "confidence" in Figure 4
>
> > it's a bit unclear what is meant with 'more confident' - I think it is probably meant that it is less blurry? I'd be great to mention this explicitly if this is the case. Also, it'd be great to somehow quantify these findings: what are the likelihoods on the hold-out test data in the different settings? Maybe in this case (MNIST), also image-based evaluations could be performed (e.g. FID).
>
> Yes, this is a good point.  We should have clarified that by "more confident", we meant that at a given percentile of the forecast distribution (e.g., p50% = 50% confidence), the bottom parts of the digit have more non-zero values, implying the networks have more certainty in the digit being generated.  While we compare model performance on MNIST using standard forecasting metrics in Table 1, we like the idea of computing likelihood and image-based evaluations for this dataset, and will include these in the final paper.
>
> ### Regarding Figure 6
>
> > "per 100" in the figure itself. I think it'd be better to re-do the Figure with "inference time per forecast" and then put it there directly - otherwise it lacks a unit.
>
> Agreed.  We will change this.
>
> ### Regarding prior probabilistic forecasting models
>
> > There is some prior work on multi-variate forecasting that has some similarity to this paper, but hasn't been mentioned. E.g. [1] (please also check reference & citations of this work), though focussing on the full multi-variate time-series problem, do also use an 'inner' architecture to generate data, and then have an outer autoregressive model. These should be mentioned in the related literature.
>
> Also a good point.  In response to your comment and also suggestions from Reviewer 6VCF, we will revise the related work to add an initial subsection on "Probabilistic autoregressive forecasting", and we will note that SutraNets are fully *compatible* with prior techniques to estimate/sample an output probability distribution at each step. For example, SutraNets can serve as the RNN model both with the AR denoising diffusion model - as box RNN in Figure 1 in [Rasul et al., 2021a](https://arxiv.org/abs/2101.12072), or with the AR conditioned normalizing flows - as box RNN in Figure 1 in [Rasul et al., 2021b](https://arxiv.org/abs/2002.06103). While these papers focus on short-term multivariate forecasting, there is no reason these estimators could not also be applied to long-term forecasting, where SutraNets would likely prove very helpful as the sequential backbone.
>
> Moreover, while our paper shows that SutraNets improve the accuracy of C2FAR ([Bergsma et al., 2022](https://openreview.net/forum?id=lHuPdoHBxbg)), we should definitely have clarified that C2FAR is a state-of-the-art distribution estimator, and was previously shown to itself improve over DeepAR ([Salinas et al., 2020](https://www.sciencedirect.com/science/article/pii/S0169207019301888)), DeepAR-binned ([Rabanser et al., 2020](https://arxiv.org/abs/2005.10111)), SQF-RNN ([Gasthaus et al., 2019](https://proceedings.mlr.press/v89/gasthaus19a.html)), and IQN-RNN ([Gouttes et al., 2021](https://arxiv.org/abs/2107.03743)) - all essentially different methods for distribution estimation that operate on top of a standard sequence model. In our paper's evaluation, we went deep in testing C2FAR alongside various enhancements to the core sequence model, such as adding lags, dropout, and frequency hierarchies. However, it would certainly help our case to go broader and show that SutraNets also enhance other methods for distribution estimation, and we will pursue such experiments in advance of the camera-ready deadline.
>
> ### Regarding conditioning on covariates
>
> > in eq.2 the conditioning is on $x^k_{1:T+N}$, i.e. $y_{\\tilde{t}}$ is conditioned on co-variates from the future. I assume this should be $x^k_{1:\\tilde{t}}$ instead?
>
> Yes, we should clarify this.  Since covariates are known *a priori*, mathematically there is no reason not to condition on covariates over all timesteps.  However, in prior forecasting models such as DeepAR ([Salinas et al., 2020](https://www.sciencedirect.com/science/article/pii/S0169207019301888), see Eq. (2)), it is common to define the RNN as consuming the current covariates alongside the prior observation at each step, so the output is implicitly only conditioned on covariates at $\\le t$, which, as you have noted, is also what we do (see our Eq. (3) and definition of RNN in line 131).
>
> FYI, in practice, ignoring future covariates is not really a limitation, as we can always introduce new covariates at time $t$ that identify future covariates, e.g., they may say something like, "there is a sale on this product beginning at time $t+1$".  In prior work, most covariates are either static across the whole prediction range (e.g., a product ID), or provide timestep-specific information (e.g., the hour-of-the-day or day-of-the-week of that particular timestep).
>
> We will revise the paper to make it clear when we are dropping conditioning on all covariates, to condition on only those up to time $\\tilde{t}$, and explain the rationale for this as noted above.  See also our similar response to Reviewer 7ywN on this topic.

---

> > ### Comment · Reviewer_BQ6D · 2023-08-14
> > **Reply**
> >
> > I thank the authors for their diligent efforts in addressing the concerns raised in my initial review.
> >
> > The minor concerns I had initially identified have been adequately discussed in the authors' rebuttal. At this point, I do not feel to have further points to discuss with the authors.

---

### Official Review · Reviewer_7ywN · 2023-07-06

**Soundness:** 3 good
**Presentation:** 3 good
**Contribution:** 3 good
**Rating:** 6
**Confidence:** 5

**Summary:**

The authors propose new ways execute the recurrence in auto-regressive  forecasting models. The approach is practical and shouldn't be too complicated to implement for a forecasting practitioner. Experiments show performance uplifts on common real and toy datasets.


**Strengths:**

The approach is interesting and practical, although the motivation appear a bit heuristically. One important advantage that is mentioned but not emphasized much, is that due to the lagged recurrence, roughly $K$-fold parallel processing can be achieved during training, which can be rather significant for RNN models on long sequences.

The paper is well organized, clearly written and easy to follow. The background section is nice and informative and provides a good overview over related work.

**Weaknesses:**

The authors explore different interleaved recurrence orders, but there is more to the topic that could strengthen the paper. For example, how can the performance be attributed  to different recurrence order on the one hand, and different hidden state dynamics on the other hand? As an example Fig 2(a) and (c) have the same recurrence order, but different hidden state dynamics. Accordingly  Backfill-alt Fig 2(f) could perhaps perform even better with the straight hidden state dynamics of a standard RNN.

Another analysis I'd love to see is how the conditional distributions $p(y_t| ...)$ look like for the different methods. Is there a recurrence order where these are mostly Gaussian or have another preferable simple shape? Take for example $y_1\sim\mathcal N(0,\sigma_1)$ and $y_2\sim\mathcal N(y_1,\sigma_2)$, then both $p(y_1)$ and $p(y_2|y_1)$ are Gaussian, but for $p(y_2)$ and $p(y_1|y_2)$  the later has a complicated bimodal form. Similar effects may impact the performance of auto regressive models in general, and it would be interesting to see that analyzed for the proposed models.

**Questions:**

* The equation (2) is incorrect:  the right hand side terms must condition on $\mathbf x_{1:T+N}$ not $\mathbf x^k_{1:\tilde T+\tilde N}$ in order for both sides to be equal. That has implications for the RNN formulation in l131 where in the current form the RNN state $h^k_\tilde t$ does not depend on any $\mathbf x_t$ with $t>K\tilde t$ effectively, while in equation (1) it does.

* Equation (2) and the inline formula in l131 describe the Regular-alt case. It would be helpful to have these explicitly formulated also for the other 3 cases, so please add them. That could save space between l135 and l146

* l143-145: *"When not conditioning on > k values, SutraNets can instead generate the complete prediction range of the kth sub-series before generating any predictions for > k ones"*. I think that's not correct. When sampling according to xxx-non schema, the RNN has to be conditioned on $h_{\tilde T+\tilde N}^{<k}$ in order to get (wrapped up) information about the whole past sub-series $y_{\tilde T:\tilde T+\tilde N}^{<k}$ and not just $y_{\tilde t}^{<k}$. This may have implications to the experiments and currently impose an unfair disadvantage to the xxx-non methods.

**Limitations:**

limitations are properly addressed

---

> ### Author Rebuttal · Authors · 2023-08-09
>
> Thank you so much for your detailed and constructive review, and your kind words regarding the advantages of the approach and the paper's organization, clarity, and overview of related work.
>
> ### Regarding parallelization of training
>
> >  One important advantage that is mentioned but not emphasized much, is ... roughly $K$-fold parallel processing can be achieved during training, which can be rather significant for RNN models on long sequences
>
> This is a great point: we will definitely emphasize this advantage in the intro, and contrast SutraNets to prior work along these lines. For example, see the proposed new summary table (Table 0) in our response to Reviewer 6VCF.
>
> ### Regarding disentangling the effects of generative order vs. hidden state dynamics
>
> > Backfill-alt Fig 2(f) could perhaps perform even better with the straight hidden state dynamics of a standard RNN
>
> So basically have a single evolving RNN hidden state, but step through the time series in backfill order in sub-sequences of $K$ consecutive values (akin to reading a document downwards but from right-to-left on each line). Essentially, a version of 2(f) where the state arrows always connect the output in row $i$ to the output in row $i+1$, rather than connecting states within sub-series. Since signal path would be high, and error accumulation lower, we agree this would help disentangle the effects of generative order vs. hidden state dynamics. Also, what is great is that this simply requires a kind of re-shuffling of the values in the conditioning and generation windows (i.e., a simple pre-processing step) before applying any standard sequence model. We will definitely pursue this further - great idea!
>
> ### Regarding visualizing the conditional distributions for the different methods
>
> > Another analysis I'd love to see is how the conditional distributions look like for the different methods
>
> This is also a good suggestion, as it may help shed light on *why* certain orderings work better. We have the machinery to plot the distributions, we just need to think further about how to systematically and objectively analyze these plots.
>
> ### Regarding conditioning on covariates
>
> > The equation (2) is incorrect: the right hand side terms must condition on $x_{1:T+N}$ not $x^k_{1:\\tilde{T}+\\tilde{N}}$ in order for both sides to be equal. That has implications for the RNN formulation in l131
>
> Yes, good catch: here we break strict equality. While mathematically there is no reason not to condition on *all* the covariates, in prior forecasting models such as DeepAR ([Salinas et al., 2020](https://www.sciencedirect.com/science/article/pii/S0169207019301888), see Eq. (2)), it is common for the RNN to consume the current covariates alongside the prior observation at each step, so the output is implicitly only conditioned on covariates at $\\le t$. We will fix this by conditioning on all the covariates in Eq. (2) (with strict equality), and then using approximate equality in Eq. (3) when we introduce the RNN, which consumes sub-series covariates. We will also provide more explanation.
>
> Regarding the implications for the covariates input to each sub-series RNN: note that in practice, ignoring future covariates is not really a limitation, as we can always introduce new covariates at time $t$ that identify future covariates, e.g., they may say something like, "there is a sale on this product beginning at time $t+1$". Moreover, in prior work, most covariates are either static across the whole prediction range (e.g., a product ID), or provide timestep-specific information (e.g., the hour-of-the-day or day-of-the-week of that particular timestep).
>
> ### Regarding full equations for all generative orderings
>
> > Equation (2) and the inline formula in l131 describe the Regular-alt case. It would be helpful to have these explicitly formulated also for the other 3 cases
>
> Yes, we can do this. For the benefit of other reviewers, we note the non-alternating formulas arise when removing the purple $y^{>k}_{1:\\tilde{t}-1}$ as a conditioning value.
>
> ### Regarding non-alternating models
>
> > "When not conditioning on $>k$ values, SutraNets can instead generate the complete prediction range of the $k$th sub-series before generating any predictions for $>k$ ones". I think that's not correct. When sampling according to xxx-non schema, the RNN has to be conditioned on $h_{\\tilde{T}+\\tilde{N}}^{<k}$ in order to get (wrapped up) information about the whole past sub-series ... This may have implications to the experiments and currently impose an unfair disadvantage to the xxx-non methods.
>
> We should clarify that the $k$th sub-series RNN does not condition on *hidden states* from other sub-series RNNs, rather only on *values* from other sub-series, which the $k$th sub-series RNN consumes at each timestep along with its own prior values, as illustrated in Figure 2. Each sub-series RNN thus gets "wrapped up" information about the other sub-series in its own hidden state, but only *up to the current timestep* - and, for non-alternating models, only for sub-series with index $<k$. Furthermore, in Figure 2(d), note that when generating y5 in row 7, we *could* theoretically condition on y6 and y7 at this point, but we do not. In practice, this seems to impair the performance of Regular-non (as noted in Footnote 3 on page 7), but not Backfill-non, where all the proximal values (before and after the value being generated) *are* conditioned on (input either at the current or at the previous timestep) - we will expand Footnote 3 to explicitly contrast the regular-non and backfill-non situations.
>
> Now, it is not clear that ignoring distant future values is a particular disadvantage to the non-alternating models, since the alternating models (and standard RNNs) also only condition on values generated up to and including the current timestep, however it would be informative to us if you see it differently.

---

> > ### Comment · Reviewer_7ywN · 2023-08-18
> >
> > Thanks to the authors for their rebuttal. I think my concerns where addressed and suggestions followed, but no major step changes were made to the paper.  My initial rating seems to resonate with the other reviewers, so I'm gonna stick with it.

---

### Official Review · Reviewer_MDp2 · 2023-07-06

**Soundness:** 3 good
**Presentation:** 4 excellent
**Contribution:** 3 good
**Rating:** 6
**Confidence:** 4

**Summary:**

The author proposed SutraNets, a novel method for neural probabilistic forecasting of long-sequence time series. It addresses challenges faced by previous autoregressive approaches, such as error accumulation and modeling long-distance dependencies. SutraNets treat long predictions as multivariate predictions over lower-frequency sub-series, reducing errors and signal path distances. The authors conduct extensive experimental results on real-world datasets, and they show significant improvements.

**Strengths:**

1. The sub-series in long-sequence analysis have seldom been investigated, especially for RNN-type networks.
2. The paper is well-organized, and the discussion is very clear.

**Weaknesses:**

1. What is the deviation point of high and low frequency? What is the connection between traditional signal analysis?
2. The performance improvement seems moderate. Could the author include std for complete?
3. Finding 4 could include more recent SOTA methods.

**Questions:**

1. Could the author discuss more about this work compared with pathTST (2023)?
2. See above.

**Limitations:**

Yes

---

> ### Author Rebuttal · Authors · 2023-08-09
>
> Thank you for your very useful comments, and for your kind words regarding the paper's key idea, organization, and clarity.
>
> ### Regarding selection of high and low frequencies
>
> > What is the deviation point of high and low frequency? What is the connection between traditional signal analysis?
>
> This is a good point: we did not describe how we chose the frequencies.  While the high frequency is implicitly the sampling frequency of the time-series, the low frequency (both for SutraNets and for the Low2HighFreq baseline) follows from the setting of $K$, which we basically viewed as a kind of hyperparameter to be set "heuristically" (line 201).  We will clarify in the methods section that $K$ is typically chosen such that it divides evenly into the primary seasonal period (i.e., the low frequency is twice or four times per day for series with primarily daily seasonality).  We will also gratefully use your idea and note in the paper that techniques such as autocorrelation and Fourier analysis can reveal the primary seasonality, and hence inform the choice of $K$, in cases where such information is not known *a priori*.
>
> Note also we still found a strong benefit of SutraNets over baselines even when $K$ does not divide evenly into the seasonal period, particularly for the alternating generative order.  See Finding 5 (line 283) and Table 3 for more details.
>
> ### Regarding error bars for experimental results
>
> > The performance improvement seems moderate. Could the author include std for complete?
>
> Yes, note we do study the (very consistent) stability of improvement in supplemental section B.2, in particular supplemental Figures 3 and 4, and we noted the overall finding in easy-to-miss Footnote 2 of the main paper.  We should promote Footnote 2 to be part of the main paper discussion and expand on our findings there.
>
> ### Regarding recent "deeper" SOTA methods
>
> > Finding 4 [Deeper models enable improved long-sequence forecasting] could include more recent SOTA methods.
>
> This is a very good point.  Beyond strengthening our finding, it would be of significant practical and theoretical importance to know if handling of long-term information can be improved in recent SOTA methods by simply increasing the depth of the underlying neural network.  We will definitely pursue this direction, thanks!
>
> ### Regarding relationship to PatchTST
>
> > Could the author discuss more about this work compared with pathTST (2023)?
>
> Yes, this paper is definitely worth adding to our sub-section "Modeling long-term dependencies" (and to a proposed new summary table, see Table 0 in our response to Reviewer 6VCF).  We will discuss PatchTST as follows:
>
> > In contrast to SutraNets, which operate over sub-series of the original input (i.e., points spaced $K$ apart), PatchTST groups $K$ *consecutive* values into input *patches*.  While patching was proposed as a method to reduce the attentional complexity of *Transformers*, patching could also be used with RNNs, where it would effectively reduce the signal path by a factor of $K$ (taking in $K$ inputs each step, with $K$ times fewer inputs overall).  However, PatchTST does not provide a mechanism to probabilistically *generate* patches, rather it directly outputs non-patched point predictions at all timesteps in one shot.

---

### Official Review · Reviewer_6VCF · 2023-07-07

**Soundness:** 2 fair
**Presentation:** 3 good
**Contribution:** 2 fair
**Rating:** 4
**Confidence:** 5

**Summary:**

This manuscript proposes SutraNets for long-range probabilistic forecasting on time series data and pixel sequences.
SutraNets is a type of recurrent neural network that transforms a long series into a collection of shorter sub-series. Sub-series forecasts are generated autoregressively, sequentially conditional on each other, enabling coherent outputs.


**Strengths:**

Figure 2 presents the generation ordering in SutraNets well, making it easy to follow.



**Weaknesses:**

Since this work focuses on probabilistic forecasting, it should include related work and baselines on time series probabilistic forecasting models.

However, most related baselines are missing. For example:
- NeurIPS’21 Probabilistic Transformer for Time Series Analysis
- ICLR’21 Multivariate Probabilistic Time Series Forecasting via Conditioned Normalizing Flows
- ICML’21 Autoregressive Denoising Diffusion Models for Multivariate Probabilistic Time Series Forecasting

“long-sequence” in the title means to predict a long sequence, it would be better to compare recent long-range time series prediction models. This is because the main work in this paper is the designed recurrent structure, rather than a new way to estimate the uncertainty. In other words, the connection between the proposed model and the probabilistic modeling is weak.

Moreover, I believe that the structure of SutraNets is quite similar to recurrent neural networks with skip or residual connections in the temporal direction. And it does not depend on the K. It is unclear what advantages do SutraNets have over RNNs with skip connections. The main differences between SutraNets and the following works should be discussed.
- NIPS’17 Dilated Recurrent Neural Networks
- NIPS’16 Architectural Complexity Measures of Recurrent Neural Networks

**Questions:**

see weaknesses

**Limitations:**

Limitations and broader impacts have been discussed in Section 3.

---

> ### Author Rebuttal · Authors · 2023-08-09
>
> Thank you very much for your helpful and insightful review. Your pointers to related work will enable us to substantially improve the paper. It is also gratifying to know that Figure 2 was valuable and contributed to the overall good presentation.
>
> ### Regarding prior probabilistic forecasting models
>
> > Since this work focuses on probabilistic forecasting, it should include related work and baselines on time series probabilistic forecasting
>
> Please refer to the section "Regarding prior probabilistic forecasting models" in our rebuttal to Reviewer BQ6D for full details on how we will use your suggestions to address this.
>
> ### Regarding [Tang & Matteson, 2021](https://proceedings.neurips.cc/paper/2021/hash/c68bd9055776bf38d8fc43c0ed283678-Abstract.html): Probabilistic Transformer for Time Series Analysis
>
> This approach is worth mentioning in our Section 2 sub-section "Reducing training/inference discrepancy", as it is non-autoregressive, but still probabilistic. However, since the predicted outputs do not condition on each other (see their Eq. 2), the output values would not be generated coherently. Without modeling local dependencies among outputs, the model can only provide a very weak assessment of whether a given sequence is likely, e.g., for anomaly detection, interpolating missing values, etc. As mentioned, our goals are more aligned with flexible autoregressive models like GPT3, where not modeling local dependencies would result in, e.g., ChatGPT generating incoherent responses.
>
> ### Regarding the connection to probabilistic modeling
>
> > the main work in this paper is the designed recurrent structure, rather than a new way to estimate the uncertainty... the connection between the proposed model and probabilistic modeling is weak
>
> We will clarify in our intro that, in our use case, quantifying the variance in a predicted output is not a "nice-to-have", but essential for downstream decision makers that use our predictions to make risk-aware resource allocation and capacity planning decisions. Computing a full likelihood of a long time series also facilitates other important use cases such as anomaly detection, missing value interpolation, etc. We thus view SutraNets as fundamentally a novel way to compute a joint probability of very long time series forecasts, as opposed to an RNN-specific enhancement. Indeed, you can apply the SutraNet factorization to both LSTMs and Transformers (see our response to reviewer iCD8). While better capturing long-term dependencies *is* one of the advantages of SutraNets in RNNs (and could be applied more broadly), it goes along with reducing error accumulation in autoregression, enabling a $K$-fold improvement in training parallelism, and enabling coherent outputs - advantages that are simultaneously lacking in other proposed probabilistic models, as we shall discuss next.
>
> ### Regarding advantages of SutraNets over other RNNs
>
> > It is unclear what advantages do SutraNets have over RNNs with skip connections
>
> This is valuable feedback. We will unify the discussion of lagged/residual inputs (lines 84-85), and skip connections between LSTM states (categorized as multi-rate recurrence layers in lines 64-65). We will also add a summary section to the related work, which will include the following new table:
>
> | RNN method | Meaning of K | RNN signal path | Max generative stride without feedback | RNN training parallelism | Coherent outputs |
> |-|-|-|-|-|-|
> | Standard RNNs, e.g. LSTMs ([Hochreiter & Schmidhuber, 1997](https://dl.acm.org/doi/10.1162/neco.1997.9.8.1735)) | N/A | $\\mathcal{O}(N)$ | $\\mathcal{O}(1)$ | $\\mathcal{O}(1)$ | Yes |
> | Skip connections ([Zhang et al., 2016](https://proceedings.neurips.cc/paper/2016/hash/860320be12a1c050cd7731794e231bd3-Abstract.html)), lagged/residual inputs ([He et al., 2016](https://arxiv.org/abs/1512.03385)) | Skip/lag amount | $\\mathcal{O}(N/K)$ | $\\mathcal{O}(1)$ | $\\mathcal{O}(1)$ | Yes |
> | PatchTST (as RNN) ([Nie et al., 2023](https://arxiv.org/abs/2211.14730)) | Size/stride of patches | $\\mathcal{O}(N/K)$ | N/A | $\\mathcal{O}(1)$ | No |
> | DilatedRNN ([Chang et al., 2017](https://arxiv.org/abs/1710.02224)) (dilations from $C \\ldots K$) | Highest dilation amount | $\mathcal{O}(N/K)$ | $\\mathcal{O}(C)$ | $\\mathcal{O}(C)$ | Only if $C=1$ |
> | SutraNets | Number of sub-series | $\\mathcal{O}(N/K)$ | $\\mathcal{O}(K)$ | $\\mathcal{O}(K)$ | Yes |
>
> > Table 0: Advantages of SutraNets over prior RNNs: prior methods can reduce signal path, but not all methods improve error feedback nor facilitate training parallelism. Patching, as used in PatchTST, can reduce signal path by grouping consecutive elements into "patches", but it does not provide probabilistic outputs. Dilated RNNs with minimum dilation amounts $>1$ do facilitate error reduction and parallel training, but at the cost of sacrificing coherency, as the model is then "equivalent to multiple shared-weight networks, each working on partial inputs" ([Chang et al., 2017, Section 4.4](https://arxiv.org/abs/1710.02224)). SutraNets reduce signal path distances while simultaneously reducing error accumulation, enabling training parallelism, and generating coherent outputs. Experimentally, SutraNets perform better than standard LSTMs and LSTMs with lagged/residual inputs.
>
> We will also add a discussion of Dilated RNN to sub-section "Modeling long-term dependencies", where we will provide the example of a time series that spikes to a high value at the very final historical (conditioning) input. In a Dilated RNN (with $C>1$), some of the outputs will be generated conditional on this spike, while others will be generated completely independently of the spike, and independent of other outputs that were aware of the spike, resulting in highly-incoherent output. The Dilated RNN paper mitigates this for point predictions by adding a final "fusion layer", but such an approach is not compatible with probabilistic forecasts.

---

> > ### Comment · Reviewer_6VCF · 2023-08-17
> >
> > Thank you for your response.
> >
> > After carefully reading your response and re-evaluating the manuscript, I still have the following questions:
> >
> > I believe the main contribution of this work lies in the long-sequence decoding strategy (Backfill-alt is suggested) for autoregressive models, which aims to reduce error accumulation (mentioned in the abstract).
> >
> > However, it is important to note that this strategy is still limited.
> >
> > For instance, in Figure 1, the approach uses (y0, y3, y6, y9) to generate (y12, y15, y18) initially, and then employs (y12, y15, y18) along with the history (y1, y4, y7, y10) to generate (y13, y16, y19).
> >
> > This raises some questions:
> >
> > - I noticed that y12, y15, and y18 are directly included in the final output, even though they were generated using only (y0, y3, y6, and y9) without considering the recent values y10 and y11. **If (y12, y15, y18) turn out to be incorrect predictions, would these errors accumulate and affect the subsequent predictions of (y13, y16, y19)?**
> > - Why wasn't (y10, y11) used to generate y12? It seems that predicting y12 using (y10, y11) would be much easier (short-term prediction).
> > - Moreover, since y12 is decoded using y0, y3, y6, and y9, **does it imply that the model assumes the time series to be periodic**? (The time series depicted in Figure 3 appears to be relatively straightforward to predict using existing long-term prediction methods.)
> >
> > There is a related work TimesNet [1], which shares a similar idea to SutraNets. Both approaches transform long time series into multiple subseries. Considering this, I have another concern regarding **why the authors did not compare their proposed model with recent long-time series prediction methods such as [1]**. This question was not addressed in their response.
> >
> > [1] [Timesnet: Temporal 2d-variation modeling for general time series analysis](https://arxiv.org/abs/2210.02186)
> >
> > I believe it would be relatively straightforward to adapt long-time prediction methods for probabilistic forecasting or to evaluate SutraNets in the context of non-probabilistic forecasting. This would be valuable in illustrating the prowess of SutraNet in long-sequence prediction.
> >
> > Given the limited evaluation presented and the absence of baselines for long-time series prediction methods, I remain uncertain about the true effectiveness of the proposed model in **long sequence** prediction.

---

> > > ### Author Response · Authors · 2023-08-18
> > > **Reply to Official Comment by Reviewer 6VCF (1/2)**
> > >
> > > We really appreciate that you have both helped improve the paper, and are now re-evaluating the manuscript in light of our work together.
> > >
> > > > the main contribution of this work lies in the long-sequence decoding strategy ... which aims to reduce error accumulation
> > >
> > > Based on reviewer feedback, we will definitely revise the abstract to emphasize the novel benefit of SutraNets is the ability to simultaneously: (1) capture long-term dependencies by reducing signal path, (2) reduce error accumulation in autoregression, (3) enable a K-fold improvement in training parallelism for RNNs, and (4) generate coherent outputs.
> > >
> > > > in Figure 1, the approach uses (y0, y3, y6, y9) to generate (y12, y15, y18) initially, and then employs (y12, y15, y18) along with the history (y1, y4, y7, y10) to generate (y13, y16, y19)
> > >
> > > We understand you to be extrapolating the process in Fig. 1(d), for **Regular-non**. Indeed, it is a *non*-alternating model: the first sub-series prediction is generated end-to-end while only using historical values for that subseries.
> > >
> > > > If (y12, y15, y18) turn out to be incorrect predictions, would these errors accumulate and affect the subsequent predictions of (y13, y16, y19)?
> > >
> > > Errors do still accumulate, but to a lesser extent. Consider generating y19. In a standard RNN (Fig. 1a), we must generate y11, y12, y13, y14, y15, y16, y17, y18 - 8 steps - in order to generate y18, i.e., the value that precedes y19. However, in Regular-non, we can generate y13, 15, y18 - 3 steps - in order to generate y18. So the prediction of y19 can be based on proximal values (y18) themselves generated with $K$X less error accumulation.
> > >
> > > > Why wasn't (y10, y11) used to generate y12? It seems predicting y12 using (y10, y11) would be much easier
> > >
> > > Regular-non by definition generates y12 using y9 and earlier, while standard RNNs (Fig. 1a) and Regular-alt (Fig. 1c) generate sequentially as you are suggesting. Yet interestingly, Regular-non sometimes works better than Regular-alt in experiments. Why? Well, During *training*, sequential RNNs sometimes find it effective to just repeat the previous gold/true value, i.e., to predict y12 based on gold y11. But in *inference*, if y11 is generated after a long chain of error accumulation, repeating y11 to predict y12 is problematic. By forcing the network to attend to values $K$ steps in the past, we reduce train/test "discrepancy" and improve forecasting accuracy.
> > >
> > > > since y12 is decoded using y0, y3, y6, and y9, does it imply that the model assumes the time series to be periodic?
> > >
> > > SutraNets do not require data to be periodic to be effective, e.g., dataset `mnist`$^{\pi}$ comprises images after a fixed random permutation, and `azure` and `wiki` comprise many trending but non-seasonal series. However, for time series that *are* periodic, we asked a similar question to yours: does it matter whether y0, y3, y6, etc. captures seasonality, i.e., does it matter whether $K$ divides into the seasonal period?  The results are in Table 3 and Finding 5: non-alternating models are comparatively weaker than alternating versions.
> > >
> > > > it is important to note that this strategy is still limited
> > >
> > > While *non-alternating* RNNs generate future values without "seeing" all the history, keep in mind that **all the alternating RNNs do see all the history in their generative process**. That is, the orange RNN in Fig. 1(f) will have seen y0, y1, y2, ..., y7, y8 when it goes to generate y11. True, it does not see y9 or y10, but that's the point: all values from y9 onward must be generated, and Backfill-alt generates y11, then y10, then y9. Since this seems to work quite well, we would like to share these findings with the community.
> > >
> > > > There is a related work TimesNet [1], which shares a similar idea to SutraNets. Both approaches transform long time series into multiple subseries
> > >
> > > We can certainly include discussion of TimesNet. It is interesting that TimesNet transforms univariate time series into 2D tensors of *multiple* periods, which are then processed via Inception-style 2D kernels (i.e., now using a 2D computer vision backbone, and outputting point predictions). SutraNets on the other hand transform time series into $K$ sub-series of the *same* period, and proposes an autoregressive factorization (with different generative orderings) for generating these sub-series using their own and previous values (with a 1D sequential generative model that produces probabilistic outputs).

---

> > > > ### Author Response · Authors · 2023-08-18
> > > > **Reply to Official Comment by Reviewer 6VCF (2/2)**
> > > >
> > > > > why the authors did not compare their proposed model with recent long-time series prediction methods such as [1]
> > > >
> > > > Experimentally, we compare SutraNets to other methods that reduce signal path and mitigate error accumulation, while generating coherent probabilistic outputs. For experimental rigour, we only selected methods that we could fairly re-implement in our codebase, so that all methods share the same normalization (based on the conditioning window), optimizer, sampling of training data, decaying learning rates, criteria for early stopping, fixed and tuned hyperparameters, etc.
> > > > - We implemented C2FAR (baseline), C2FAR+lags (to improve signal path), C2FAR+dropout (to mitigate error accumulation), Low2HighFreq (to leverage low-frequency predictions to guide high-frequency generation, similar to Scaleformer), plus four SutraNet variations.
> > > >
> > > > Ideally we could also compare to the performance of other implementations on standard datasets. Unfortunately, **comparing to the evaluation in the TimesNet paper (and other recent work) is problematic**.
> > > >
> > > > - As explained to Reviewer iCD8 (regarding Scaleformer), TimesNet (like Scaleformer, Autoformer, and FEDformer) is restricted to only using at most 96 historical values, i.e., only **4 days** of hourly data (Table 2: "the past sequence length is set as 36 for ILI and 96 for the others"). Note we use up to 2016 historical values -- **21X the maximum lookback context used in TimesNet** -- so TimesNet and SutraNets are not really comparable in terms of the scale of historical data considered.
> > > >
> > > > - More importantly, the `traffic` dataset (and to a lesser extent, `electricity`) has extremely strong **weekly** seasonality - i.e., a future value is accurately predicted by the value 168 steps (hours) in the past. Since the implemented TimesNet cannot "see" values from one week ago, it is likely to perform much worse than even the seasonal-naive-1week baseline (which itself performs worse than SutraNets). We strongly advocate including the seasonal-naive baseline in all forecasting results, as it facilitates comparison. In the words of [Hewamalage et al., 2023](https://arxiv.org/abs/2203.10716): "on series that have clear seasonal patterns, models should accordingly be benchmarked against the seasonal naive model."
> > > >
> > > > - That being said, please see our recent response to Reviewer iCD8, where we now compare our proposed model to the recent long-time series prediction method of Scaleformer, based on published Scaleformer results on `traffic` and `electricity`. Unlike with TimesNet, this comparison is facilitated by Scaleformer generating a probabilistic forecast, and evaluating using comparable metrics to us (albeit still with restricted context).
> > > >
> > > > > it would be relatively straightforward to adapt long-time prediction methods for probabilistic forecasting or to evaluate SutraNets in the context of non-probabilistic forecasting. This would be valuable in illustrating the prowess of SutraNet in long-sequence prediction
> > > >
> > > > Indeed, it would be quite impressive if a system designed for probabilistic forecasting was competitive with the accuracy of all the non-probabilistic forecasting systems on that task, and yet we have reason to believe that this is the case: most recent "long sequence" systems will be negatively impacted by only using limited context windows, as described above. However, while this would be nice to show definitively, it's still an apples-to-oranges comparison. That is, and I am sure you would agree, it need not be a *requirement* for publication that systems designed for probabilistic forecasting also establish competitive results on non-probabilistic forecasting.
> > > >
> > > > > Given the limited evaluation presented and the absence of baselines for long-time series prediction methods, I remain uncertain about the true effectiveness of the proposed model in long sequence prediction
> > > >
> > > > While your point is well taken regarding SutraNet evaluation on the related task of *non-probabilistic* forecasting, we do provide "extensive experimental results on real-world datasets" (as noted by Reviewer MDp2) in our evaluation of *probabilistic* systems. We evaluated:
> > > >
> > > > - 8 different neural models (4 comparison, 4 SutraNet) on 6 different datasets (Table 1)
> > > > - 3 neural models with 4 different variations in neural depth on 2 datasets (Table 2)
> > > > - 2 neural models with 3 different variations in $K$ on 2 datasets (Table 3)
> > > > - 2 neural models with 5 different variations in random seed on 2 datasets (supplemental Fig. 3)
> > > >
> > > > Each of these 100+ results required running a hyperparameter tuning grid search, so, as you can imagine, the total GPU compute time was rather significant for this project, but necessary for the depth and breadth of analysis that we provide in the paper.

---

### Official Review · Reviewer_iCD8 · 2023-07-07

**Soundness:** 3 good
**Presentation:** 2 fair
**Contribution:** 2 fair
**Rating:** 5
**Confidence:** 4

**Summary:**

This paper proposes a novel method for probabilistic forecasting of long-sequence time series. It uses an autoregressive generative model to factorize the likelihood of long sequences into products of conditional probabilities.  The proposed model SutraNets treat long and univariate prediction as multivariate prediction over lower-frequency sub-series to effectively reduce error accumulation. Experimental results show improved forecasting accuracy on a variety of datasets.

**Strengths:**

1. It's novel to convert a univariate series into a multivariate series, each dimension comprising sub-series of the original sequence.

2. It's interesting that SutraNet model generates each sub-series conditional on both its own prior values and on other sub-series.

**Weaknesses:**

The proposed model is only applied to LSTM based model, while many of the state-of-the-art time series forecasting model is Transformer based. Not sure whether it can be integrated into Transformer based models.

**Questions:**

Could this approach be applied to Transformer based models?

**Limitations:**

The limitations are mentioned in Section 3. However, it doesn't show the limitations to extend it to Transformer based models which are very popular in time-series forecasting recently.

---

> ### Author Rebuttal · Authors · 2023-08-09
>
> Thank you very much for your thoughtful feedback, and your support for the paper's core idea.
>
> You raise an important point regarding Transformers: we did not spend enough time describing how SutraNets could be applied to Transformers.  While we did mention in line 45 that SutraNets could be applied to Transformers, in the methods section we only briefly noted (lines 162-163) how attending to SutraNet sub-series could lower Transformer complexity.
>
> We will add a new sub-section to Section 3:
>
> > **Application to Transformers**
>
> > SutraNets can also be applied with autoregressive Transformer models ([Li et al., 2009](https://arxiv.org/abs/1907.00235)). Rather than using $K$ separate RNNs to parameterize the conditional probabilities in Eq. (3), we can use $K$ separate Transformers.  Compared to a single standard Transformer, using Backfill and non-alternating SutraNet Transformers would have the advantage of reducing *error accumulation* (by forcing the network to predict without immediately-preceding values), but not of *signal path*, which is already $\\mathcal{O}(1)$ for standard Transformers ([Vaswani et al., 2017](https://arxiv.org/abs/1706.03762)).  Moreover, at each timestep, a SutraNet Transformer can attend over all prior values, limited only by the generative order.  In a naive implementation, alternating SutraNet models could therefore attend to essentially all prior historical values, resulting in an asymptotic complexity of $O(N^2)$ - the same as standard Transformers.  However, we can achieve $\\mathcal{O}(N^2/K)$ attentional complexity by restricting each sub-series Transformer to only attend to values from its own sub-series, plus a small number of proximal values from other sub-series (similar to strided or banded self-attention ([Child et al., 2019](https://arxiv.org/abs/1904.10509), [Brown et al., 2020](https://arxiv.org/abs/2005.14165))).  Although asymptotically larger than the $\\mathcal{O}(N^2/K^2)$ complexity of patching (Nie et al., 2023), or the $\\mathcal{O}(N (\\log N)^2)$ of LogSparse attention ([Li et al., 2009](https://arxiv.org/abs/1907.00235)), it merits further investigation to determine which approach results in the most favorable accuracy vs. complexity trade-off in practice.
>
> In the limitations section, we will also mention that SutraNets will not improve (nor harm) the parallelization of training in Transformers, since Transformers are already fully parallelizable across the sequence.
>
> Finally, we will do a better job of motivating why we evaluated SutraNets in the context of RNN-based forecasting in the first place.  Notably, our production forecasting system, like many others in industrial settings and cloud services, (e.g., via Amazon Forecast), is based on RNNs.  As our paper shows, SutraNets offer major benefits to such RNN-based systems, as SutraNets are the first approach to advance all three key dimensions of signal path, error accumulation, and parallelization of RNN training.

---

> > ### Comment · Reviewer_iCD8 · 2023-08-16
> >
> > Thanks a lot for authors on addressing my initial concerns. However, my main concern is in recent 2-3 years, latest top-tier conference, other backbone models have far more superior performance than RNN based model. Even for probabilistic forecasting, it's very easy to adapt current transformer based model in a probabilistic fashion. Please refer to [1] section 4.4 (appendix table 9) for the results with various Transformer based backbone model. Could the results on this paper beat the results shown the appendix table 9?
> >
> > [1]Shabani, Amin, et al. "Scaleformer: iterative multi-scale refining transformers for time series forecasting." ICLR2023

---

> > > ### Author Response · Authors · 2023-08-17
> > >
> > > Thank you very much for engaging in this discussion and for suggesting the insightful comparison with Scaleformer.  We absolutely do understand your concern.  While our new paragraph above clarifies that Sutranets can be applied to Transformers, it is definitely not certain that the improvements that we demonstrated will generalize to Transformers.  That being said, I am sure you would agree that NeurIPS may still include papers demonstrating advances in RNNs.  As you know, RNNs are more biologically-plausible than Transformers (maintaining and updating a memory-like state), and, unlike Transformers, which must use a fixed-size context window, RNNs can theoretically condition on context of any length.
> > >
> > > This latter point is especially relevant to comparisons with Scaleformer ([Shabani et al, 2023](https://arxiv.org/abs/2206.04038)).  In Table 9 of Shabani et al. (and in all their results), note the underlying Transformer is restricted to 96 historical values, i.e., only **4 days** of hourly data (Section 4.1: "the look-back window size is fixed to 96, and the horizon is varied from 96 to 720").  Yet also note the `traffic` dataset (and to a lesser extent, `electricity`) has extremely strong **weekly** seasonality - i.e., a future value is accurately predicted by the value 168 steps (hours) in the past.  Since the implemented Scaleformer cannot "see" these values from one week ago, it makes sense that it performs much worse than both SutraNets, *and* the seasonal-naive-1week baseline (see below).  (One may also qualitatively compare the results in their Figure 9 to our Figure 3, to see how Scaleformer may miss weekly "spikes" that SutraNets can predict).  Furthermore, note that probablistic Scaleformer uses the non-SOTA Gaussian output distribution of DeepAR (Salinas et al., 2020), while SutraNets are evaluated with the SOTA approach of C2FAR (Bergsma et al., 2023).
> > >
> > > Although Scaleformer with Transformers may be limited to short conditioning windows, the core *idea* of Scaleformer can be applied with RNNs.  Indeed, we implemented and evaluated a similar method to Scaleformer, but for RNNs (and using C2FAR as the output distribution), which we called Low2HighFreq.  We noted the connection to Scaleformer in our related work.
> > >
> > > The full comparison (repeating results from our Table 1) is summarized in the following table:
> > >
> > > | System                   | Length of conditioning | Length of prediction  | CRPS/wQL on `electricity` | CRPS/wQL on `traffic` |
> > > |--------------------------|------------------------|-----------------------|---------------|-----------|
> > > | Informer (Table 9, Shabani et al., 2023)  | 96    | 96                    | 0.330         | 0.372     |
> > > | Scaleformer  (Table 9, Shabani et al., 2023) | 96 | 96                    | 0.238         | 0.288     |
> > > | seasonal-naive-1week     | 168                    | 168                   | 0.111         | 0.175     |
> > > | Low2HighFreq (Scaleformer for RNNs) | 168         | 168                   | 0.082         | 0.166     |
> > > | SutraNets (Backfill-alt) | 168                    | 168                   | 0.074         | 0.128     |
> > >
> > > Here we *predict* 168 steps ahead vs. 96 for Scaleformer (which should disadvantage *our* results).  In advance of the camera deadline, we will repeat these evaluations to use length-96 prediction windows, and we will also train versions of our system using length-96 conditioning.  Also note Scaleformer evaluates using CRPS, while we evaluate using wQL, a 10-point approximation to CRPS (for point predictors like seasonal-naive-1week, note CRPS reduces to normalized deviation).  While in some ways it is unfair to compare systems using different amounts of context, it is nevertheless quite notable that Scaleformer, FEDformer, Autoformer, TimesNet only condition on 96 historical values for the hourly predictions in all their experiments.  We evaluated with up to length-2016 contexts, or 21X the length used in Scaleformer.  The above proposed table should provide helpful information - for both practitioners and researchers - on the potential cost of restricting the look-back context.

---

> > > > ### Comment · Reviewer_iCD8 · 2023-08-17
> > > >
> > > > Thanks a lot for the author's latest experiments. This addressed my main concern and I increased the rating to 5.

---

### Decision · Program_Chairs · 2023-09-21

**Decision:**

Accept (poster)

**Comment:**

This paper studies the problem of long and univariate prediction by treating it as multivariate prediction over lower-frequency sub-series. The proposed method reduces error accumulation. Experiments show that the proposed method improves forecasting accuracy on real datasets.

The paper received five reviews, most of which are positive, except one negative review by Reviewer 6VCF. This reviewer raised concerns about previous baselines based on RNN and Transformer, as well as connection to probabilistic modeling. These concerns were adequately addressed in the rebuttal and subsequent discussion. Another reviewer had similar concerns, but raised his or her rating to 5 after rebuttal and discussion.

Overall, the final reviews are positive about the novelty and contribution of the paper.